# Identification of high-wind features within extratropical cyclones using a probabilistic random forest - Part 2: Climatology over Europe

Lea Eisenstein[1], Benedikt Schulz[2], Joaquim G. Pinto[1], and Peter Knippertz[1]

[1]Institute of Meteorology and Climate Research - Troposphere Research, Karlsruhe Institute of Technology, Karlsruhe, Germany
[2]Institute of Stochastics, Karlsruhe Institute of Technology, Karlsruhe, Germany

**Correspondence:** Lea Eisenstein (lea.eisenstein@kit.edu)

**Abstract.** Strong winds associated with extratropical cyclones are one of the most dangerous natural hazards in Europe. These high winds are mostly associated with five mesoscale features: the warm (conveyor belt) jet (WJ); the cold (conveyor belt) jet (CJ); cold frontal convection (CFC); strong cold-sector winds (CS); and, in some cases, the sting jet (SJ). The timing within the cyclone's life cycle, the location relative to the cyclone core and further characteristics differ between these features and, hence, likely also their associated forecast errors. In Part 1 of this study (Eisenstein et al., 2022a), we introduced the objective and flexible identification tool RAMEFI (RAndom-forest-based MEsoscale wind Feature Identification), which distinguishes between WJ, CFC and CS as well as CJ and SJ combined. RAMEFI is based on a probabilistic random forest trained on station observations of 12 storm cases over Europe. Being independent of spatial distribution, RAMEFI can also be applied to gridded data. Here, we use RAMEFI to compile a climatology over 19 extended winter seasons (October–March, 2000–2019) based on high-resolution regional reanalyses of the German Consortium for Small-scale Modelling (COSMO) model over Europe. This allows the first-ever long-term objective statistical analysis of the mesoscale wind features, their occurrence frequency, geographical distribution and characteristics. For Western and Central Europe, we demonstrate that the CS is prominent in most winter storms, while CFC is the least common cause of high winds, both in terms of frequency and affected area. However, probably due to convective momentum transport, CFC is on average the cause of the highest gusts after the CJ, and has the highest gust factor. As expected, CFC high-wind areas show high levels of humidity and overcast conditions. In contrast, CS is characterised by sunnier conditions interspersed by patchy cumulus clouds leading to a broader cloud cover distribution than for other features. The WJ produces the weakest winds on average, but affects a larger area than CJ. Central Europe is more strongly affected by WJ and CFC winds, while the CJ usually occurs farther north over the North and Baltic Seas, northern Germany, Denmark and southern Scandinavia. System-relative composites show that WJ and CFC tend to occur earlier in the cyclone life cycle than CJ and CS. Consistently, CS is the most common cause of high winds over Eastern Europe, where cyclones tend to occlude, represented by a narrowing warm sector and weakening cold front. The WJ mostly occurs within the southeastern quadrant of a cyclone bordering with the narrow CFC in the west. However, the location of CFC varies greatly between cases. The CS occurs in the southwestern quadrant, while the CJ appears closer to the cyclone centre, sometimes stretching into the southeastern quadrant. This objective climatology largely confirms previous, more subjective

investigations but puts these into climatological context. It allows a more detailed analysis of feature properties, and provides
a solid foundation for model assessment and forecast evaluation in future studies.

## 1   Introduction

High wind speeds associated with extratropical cyclones, especially during wintertime, can cause enormous damage and belong
to the most severe natural hazards (Fink et al., 2009). During its lifetime, the strongest wind gusts can be induced by different
airstreams associated with the cyclone (Hewson and Neu, 2015). As discussed in Part 1 (Eisenstein et al., 2022a) and illustrated
in their Fig. 1, strong wind gusts are mostly connected to five features: the warm jet (WJ), the cold jet (CJ), the sting jet (SJ),
cold-frontal convection (CFC) and high winds within the cold sector (CS). The WJ is part of the early stages of the warm
conveyor belt, an ascending air flow ahead of the cold front (Wernli and Davies, 1997; Eckhardt et al., 2004; Madonna et al.,
2014), while it is still near the surface it causes high winds within the warm sector of a cyclone. Here, we define CFC as the
region of high winds co-located with precipitation around the cold front, i.e., also if it is occurring ahead of the surface front
in case of forward tilted fronts. In Shapiro-Keyser cyclones (Shapiro and Keyser, 1990) the warm – or bent back – front is
usually the stronger front (Catto, 2016). Considering recent cases of Shapiro-Keyser cyclones (e.g., Egon 2017, Xavier 2017,
Friederike 2018; see Eisenstein et al., 2022a), CFC appears to be more common following the Norwegian cyclone model
(Bjerknes, 1919). Analogously to the warm conveyor belt, the CJ is associated with the cold conveyor belt ahead of the warm
front and wrapping around the cyclone centre. In contrast to the warm-conveyor belt, however, the cold conveyor belt, and
hence CJ, stays at low levels during its lifetime. The SJ is an airstream descending from mid-levels within the cloud head into
the frontal-fracture region of a Shapiro-Keyser cyclone (Clark and Gray, 2018). It can cause high wind speeds slightly ahead
of and earlier than the CJ if reaching the surface. High winds in the cold sector region, i.e., behind CFC but not associated with
the CJ or SJ, are classified as CS, which can for example include post-CFC and winds caused by dry intrusions (Raveh-Rubin
and Catto, 2019).

The RAMEFI (RAndom-forest-based MEsoscale wind Feature Identification) method introduced in Part 1 (Eisenstein et al.,
2022a) focuses on the identification of WJ, CFC, CJ, CS and high winds associated with no feature (NF). As the CJ and SJ have
similar characteristics in surface parameters due to the proximity in both time and space, the SJ is included in the more frequent
CJ feature. For the identification, the features were subjectively labelled for 12 winter storms based on surface observations,
which were then used for training of a probabilistic random forest (RF). The RF can also be applied to different data sets,
independent of horizontal resolution. It learned physical consistent characteristics such as decreasing pressure ahead of the
cold front, where the WJ is located, and increasing pressure behind the front, where CJ and CS are located. The most important
predictor for CFC is precipitation. Furthermore, the warm and cold sectors are indicated by temperature-related parameters.
Given the case-to-case variability in the parameters used here, the RF outputs a probability of feature occurrence.

Given that extra-tropical cyclones are such a dominant feature of the mid-latitudes, several objective algorithms have been
developed for identifying cyclones and their tracks from digital data, either reanalysis or climate model data (see Ulbrich et al.,
2009; Neu et al., 2013, for an overview). Depending on the perception of what a cyclone is, various variables can be used

for tracking (Hoskins and Hodges, 2002), but the most common are mean sea level pressure and 850 hPa relative vorticity. During winter, three local maxima of cyclone frequency are found over the Northern Hemisphere, namely over the North Atlantic, North Pacific and Mediterranean (Ulbrich et al., 2009). While the two first regions are identified for all methods, the maximum over the Mediterranean is dependent on the resolution of the used data set and methodology used. Neu et al. (2013) compared 15 cyclone tracking methods and found significant differences in life cycle characteristics, but a large consistency is found for long-living, intense cyclones. Dacre et al. (2012) describe the development and compilation of an extratropical cyclone atlas using 200 extreme North Atlantic cyclones over a 20-year period. The atlas includes composites of horizontal and vertical cyclone structure, multiple parameters (e.g., cloud cover, wind and relative humidity) and cyclone evolution, while also identifying the warm and cold conveyor belts and dry intrusions.

A climatology focusing on near-surface winds can be found in Laurila et al. (2021). They focus on the North Atlantic and Europe by defining an extreme wind factor, which is the monthly 98th percentile divided by the monthly mean wind speed. While they found no linear trend between 1979 and 2018, they showed that the strongest winds are mostly connected with storm tracks in the winter season. This is consistent with the review paper of Feser et al. (2015), who concluded that decadal variability is the dominant feature of storminess over the region in the last 100–150 years, and only regional and short term trends can be identified. As expected from the surface and boundary layer characteristics, Laurila et al. (2021) also identified a distinct land-sea contrast in the 10 m wind speed. The same is true for wind gusts, given for example the very different gust factors typically found for offshore / inland areas (Wieringa, 1973).

To the best of our knowledge, the first climatology focusing on different mesoscale wind regions within cyclones is Parton et al. (2010), who differentiate cold frontal events, warm-sector events, tropopause folds/warm fronts, SJs and unclassified events within data from a wind-profiling radar in Wales over a 7-year period. According to them, warm-sector events are the most common cause of strong winds with around 40 %, while cold-frontal events make up around 24 % over the investigated area, which may not be representative of cyclones in general. A study by Rivière et al. (2015) suggests that in early stages a cyclone is dominated by the WJ and later by the CJ. Hewson and Neu (2015) compiled a subjective climatology of WJs, CJs and SJs on the basis of 29 wind storms. Note that their definition of WJ also includes CFC. They created an idealised conceptual model for the timing relative to the cyclone life cycle, the location relative to the cyclone centre and their strength, while also suggesting differences in further characteristics, such as instability/stability, vertical gradient of horizontal wind speed etc. In addition to these low-level jet features, Earl et al. (2017) further distinguished various convection-induced high-wind features. Consistent with other studies, they found that WJs and CJs are most common when looking at the highest 1 % of daily maximum wind gusts, while CFC and potential SJs commonly cause the highest 0.1 % of daily maximum gusts. However, their focus lays solely on winter storms over the UK similar to Parton et al. (2010).

Recently, some first objective approaches to identify mesoscale wind features have been developed by Manning et al. (2022) and Gentile and Gray (2023). Both studies use the strong thermal (and moisture in the latter) gradient to detect fronts and define high winds on the warm side as WJ. While Manning et al. (2022) identify SJs using a kinematic objective identification and define all further high winds on the cold side of the fronts as CJ, Gentile and Gray (2023) distinguish between the CJ travelling against the system motion (named CCBa) and the CJ wrapping around the cyclone centre (CCBb) following Earl

et al. (2017). Thus, the latter resembles our definition of a CJ merged with the CS. Although Manning et al. (2022) focus on future changes and Gentile and Gray (2023) on a 9-year climatology, both works conclude that winds in the cold sector to the

west and south of the cyclone centre have higher wind speeds than in the WJ. Furthermore, Gentile and Gray (2023) analyse atmosphere-ocean-wave coupling based on ocean stations and find that the CCBb is the most common cause of high winds with an increasing proportion of CCBa to the northeast of the UK.

The goal of this study is to expand and complement existing shorter and/or more general climatologies using RAMEFI, the first tool to objectively distinguish WJ, CFC, CJ and CS. With this aim, a high-resolution regional reanalysis dataset for

19 extended European winter seasons is used. Other observational data sets are used for specific aspects or as comparison. The paper is structured as follows: First, we shortly recapitulate the data sets and method already introduced in Part 1 (Sect. 2). Section 3 focuses on the occurrence of the identified high-wind features, i.e., frequency, relative to the cyclone centre and cyclone life cycle, while Sect. 4 discusses the different characteristics of the features. Conclusions are drawn in Sect. 5.

## 2   Data and Method

Our approach is based on the novel RAMEFI method. This section briefly introduces the method and data sets used, in particular a surface observation and a gridded data set. For a full description, we refer to Part 1 (Eisenstein et al., 2022a). For the climatology, we focus on the extended winter months October to March including 19 years of data. To make sure that only winter storms are included in our climatology, cyclone tracks and further filters are used.

### 2.1   Surface observations

The observational data set provided by the German Weather Service includes hourly surface observations over land from 2001 to mid-2020 containing 5 parameters: mean sea level pressure ($p$), 2 m air temperature ($T$), wind speed at 10 m ($v$), wind direction at 10 m ($d$) and precipitation amount ($RR$). Additionally, the potential temperature ($\theta$) is computed, and $\theta$ and $v$ are normalised by their median and 98th percentile, respectively, to take the diurnal and seasonal cycles as well as location-specific characteristics into account ($\tilde{\theta}$ and $\tilde{v}$, respectively). The median and 98th percentile are computed for the specific location, time

of day and day of the year $\pm10$ days using the available time period. Furthermore, temporal tendencies of $p$, $\tilde{\theta}$ and $d$ are calculated ($\Delta p$, $\Delta\tilde{\theta}$ and $\Delta d$, respectively), here represented simply by the difference between the current and the previous hour.

As in Part 1, we concentrate on Western and Central Europe, more specifically, stations within the area of $10\,°$ W to $20\,°$ E, $40\,°$ N to $60\,°$ N.After removing stations that measure fewer than three of the five meteorological parameters, around 750

station reports per time step remain on average. For the climatology, we include all time steps from January 2001 to December 2019, i.e., a total of 114 months.

## 2.2 COSMO-REA6

COSMO-REA6 is a reanalysis data set based on the German Consortium for Small-scale Modeling (COSMO) model from the German Weather Service (DWD) computed by the Hans-Ertel-Centre for Weather Research. The data set covers the European CORDEX (Coordinated Downscaling Experiment) domain with a grid spacing of $0.055\,°$, i.e., roughly $6\,km$, and uses ERA-Interim data (Dee et al., 2011) as boundary conditions. The reanalysis is available from 1995 to mid-2019. For a fair comparison of the two data sets, the chosen time period for the climatology is as close as possible while including 19 extended winter seasons each. This means, while the observations cover January 2001 to December 2019, the COSMO-REA6 is used for October 2000 to March 2019, i.e., with a minor shift of three months. In addition to the parameters mentioned above, COSMO-REA6 allows us to include further variables, such as wind gusts at $10\,m$ ($v_{gust}$), specific humidity at $2\,m$ ($q$), relative humidity at $2\,m$ ($RH$) and total cloud cover ($cc$). The model uses a convection parametrisation by Tiedtke (1989) and wind gusts are estimated following Schulz and Heise (2003) and Schulz (2008) with a turbulent and a convective gust component. The cloud cover is based on cloud water and cloud ice and considers grid-scale, sub-grid convective and sub-grid stratiform clouds (Doms et al., 2021) and verified using ceilometer data (Bollmeyer et al., 2015). For a detailed description of the data set, we refer to Bollmeyer et al. (2015) and Doms et al. (2021). The median and 98th percentile used for the normalisation of $\theta$ and $v$, respectively, are only computed for the 10-year time period 2005 to 2015 due to computational cost. The data, originally on a rotated grid, were regridded to a latitude-longitude grid with a spacing of $0.0625\,°$, i.e., roughly $7\,km$, for the area of $10\,°\,W$ to $25\,°\,E$, $40\,°\,N$ to $65\,°\,N$. Note that the area shows an eastward extension and a northerly shift compared to the observational data set to include more northern regions affected by winter storms, where the observational data are sparse.

## 2.3 RAMEFI

RAMEFI delivers a probabilistic identification of the five features WJ, CFC, CJ, CS and NF, with SJ being included in CJ. The method is based on an RF that was trained on surface observations of 12 storm cases, for which the wind features were subjectively labelled. These case studies are picked to capture a healthy diversity of cyclone developments and features, i.e., includes very intense and more moderate cyclones with differing storm tracks. Given this diversity and the promising evaluation of the method in Part 1, we assume a reliable detection of the features in long-term data for most cyclones. Note that RAMEFI focuses on strong wind speeds, hence the RF was trained and tested only for 'windy conditions', which we define as cases with $\tilde{v} > 0.8$. In Part 1 (Eisenstein et al., 2022a), we used a cross-validation approach for a proper evaluation of the method, i.e., to test the identification for each storm, we trained the underlying RF on the remaining eleven case studies to avoid using data of the storm of interest, resulting in a total of 12 RFs. Here, however, we use the RF trained on all of the 12 case studies (Eisenstein et al., 2022b). Further, the statistical evaluation of the application on COSMO-REA6 data in Part 1 demonstrates that RAMEFI generates reliable identifications for gridded data despite being trained on surface observations. For details on the method, we refer to Eisenstein et al. (2022a).

RAMEFI is spatially independent, such that the output probability is computed individually for each station or grid point. Here, we apply RAMEFI to station observations and COSMO-REA6 data under windy conditions during the extended winter

months, regardless of whether a storm occurred or not. However, we later filter the output for cyclone occurrence as discussed in Sect. 2.4. Note that the 12 cases used for training (see Eisenstein et al., 2022a) are also included in the data used for the climatology. The reasoning behind the cross-validation approach in Part 1 was to evaluate whether the RF is able to reliably identify the features in unseen data. Here, we want to generate a climatology of the high-wind features rather than testing the method. Hence, it is unproblematic to apply RAMEFI to the same data it was trained on. Instead, we obtain an identification that mirrors the subjective identification within in these storms, and is still consistent with the entire climatology due to the same model underlying the identification.

In contrast to Part 1, RAMEFI is also applied to ocean grid points of COSMO-REA6, where it has not been systematically evaluated such that results should be treated with some caution there. Considering that the wind speed distribution over sea is broader, the 80 %-threshold of the 98th percentile of $v$ results in more windy conditions over ocean than over land (see Appendix A). Characteristics of other parameters are discussed in Sect. 4. Nevertheless, looking at various cases over the 19-year period, the ocean and land do not seem to behave fundamentally differently with respect to feature detection and their probability distributions. Exemplary cases can be accessed in the Supplement (Eisenstein et al., 2023b).

As RAMEFI provides a probabilistic identification, each feature is assigned a probability from 0 to 1. The distribution of the probabilities for each feature are shown in Fig. 1 for both data sets. While WJ and CS show a similar distribution in both data sets with peaks around 48 %, the maximum of CJ slightly differs and is lower at around 43 % for COSMO-REA6 and 40 % for observations. The highest uncertainty in the feature detection can be seen for CFC, which shows overall lower probabilities with a peak around 33 %. The biggest difference in the data sets is found for NF. COSMO-REA6 shows a peak at 50 %, but a plateau between 50 % and 73 % in the observations. This will be discussed further in Sect. 3.1. This probabilistic information is used in two different ways: Firstly, we assign the feature with the highest probability to a given time and grid point, ignoring all other probabilities (referred to as MAXP hereafter). Secondly, we exploit the probabilistic nature of the identification by interpreting the (accumulated) feature probabilities as the expected number of features (referred to as ACCP hereafter). The calibration of the feature probabilities, which was checked in Part 1 of the study, is a critical condition for this approach. The second approach is particularly important for features with less confident detection, which might be underrepresented in the first approach, e.g., the CFC as shown in Fig. 1 (see also Sect. 3.1).[1]

## 2.4 Filtering for cyclone tracks

In general, RAMEFI can be used without any filters. However, for a meaningful climatology we aim to exclude high winds not associated with extratropical cyclones and, hence, the mesoscale wind features are targeted here. So to filter the gained probabilities and also to compile a storm-relative analysis, objectively determined cyclone tracks are used. The cyclones are identified and tracked from ERA5 (Hersbach et al., 2020) $p$ data using an objective tracking algorithm (Murray and Simmonds, 1991; Pinto et al., 2005). The algorithm primarily searches for the minimum $p$ in the vicinity of a $\nabla^2 p$ maximum as cyclone centres within a radius of $750\,\mathrm{km}$. To filter out weak and thermal lows, or cyclones over high orography, we follow the criteria

---

[1]As an example, consider an identification of 75% for rain versus 25% for no rain. Although we always detect rain via the first approach, rain was actually observed, on average, every fourth case (given the probabilities are calibrated).

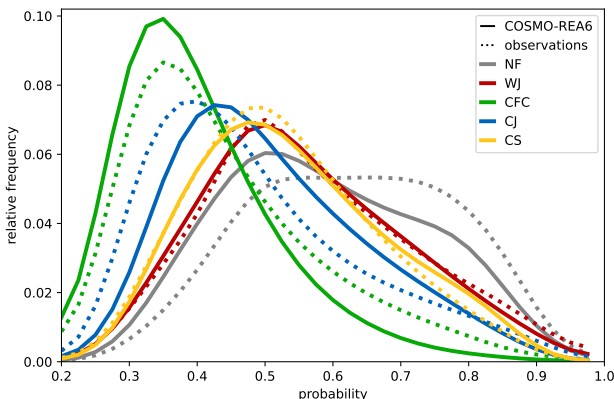

**Figure 1.** Distribution of RAMEFI probabilities for each feature over the 19 extended winter seasons using COSMO-REA6 (solid) and station observation data (dashed). The density is calculated based on smoothed histograms.

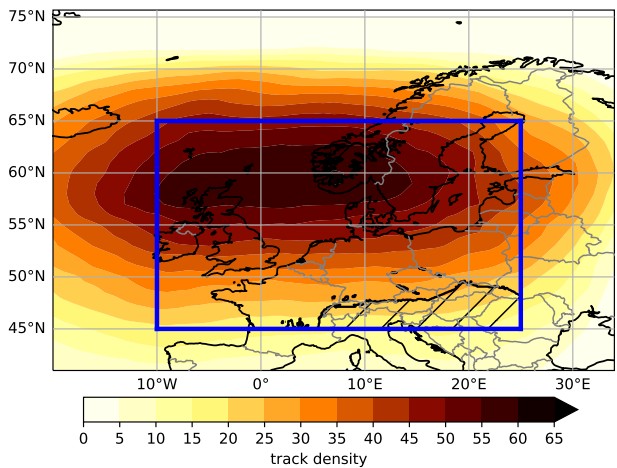

**Figure 2.** Cyclone track density in number of cyclones per year per $(^{\circ}\,\mathrm{lat})^2$ within the examined time period (2000–2019). The blue box represents the study area, with hatching indicating excluded areas.

from Pinto et al. (2009). The method settings used for ERA-Interim (Neu et al., 2013) were slightly adapted to handle the higher spatial resolution data of ERA5, while the time resolution was kept at 6 h intervals. Cyclones must travel at least 1000 km and last for at least one day to be considered. The resulting cyclone tracks are then interpolated linearly to gain hourly information. The track density, i.e., number of cyclones passing over a grid point (Ulbrich et al., 2009), of all cyclones within the 19 years can be seen in Fig. 2. As expected given the selected study area (blue box in Fig. 2), cyclone tracks corresponding to the identified features typically travel over the British Isles and the North Sea towards the Baltic Sea.

Here, all grid points showing windy conditions in the vicinity of $15^{\circ}$ in zonal direction and $-15^{\circ}$ to $5^{\circ}$ in meridional direction of the cyclone centre and the considered area are used, except those at altitudes above 800 m consistent with Part

1 (Eisenstein et al., 2022a). Furthermore, following Part 1 we exclude the Balkans, where topography is complex and where winter storms are rare. For example, in mountainous regions, the Foehn effect might be falsely identified as WJ. Hence, such areas are removed from the climatology, as indicated by the hatching in Fig. 2. We further decided to focus on the area between $10\,°$ W to $25\,°$ E, $45\,°$ N to $65\,°$ N, hence the most impacted area over Europe. Note that the changing zonal extent with latitude is neglected in this analysis. However, we do not expect this to have a significant impact on our main conclusions. We remove

time steps, where fewer than 5 % of all grid points during a time step are associated with one of the features (excluding NF), i.e., weak cyclones. Time steps with a cyclone moving through and at least 5 % of the area showing windy conditions are referred to as 'stormy time steps'. Overall, these filters lead to 1910 cyclones over around 20,000 time steps, which are included in the analysis.

## 3 Occurrence of high-wind features

One of the main aspects of this climatology is the occurrence of the mesoscale wind features in time and space. As mentioned in Sect. 1, the features develop during different times in a cyclone life cycle and in different areas of the cyclone. In addition to the overall frequency of the features as well as diurnal, seasonal, and yearly variations (Sect. 3.1), RAMEFI further gives us the possibility to obtain the occurrence of the wind features both in an Earth-relative (Sect. 3.2) and a system-relative framework, i.e., relative to the cyclone centre and relative to the cyclone life cycle (Sect. 3.3).

### 3.1 Relative occurrence frequency

Figure 3 shows the relative frequency of the wind features for both observations (first row) and COSMO-REA6 (lower three rows) over all time steps and grid points satisfying stormy time step conditions (Sect. 2.4). While the left two columns show all features, including NF, the right two columns neglect NF to focus on the identified mesoscale wind features. Furthermore, the frequencies for the most probable feature (MAXP) is displayed next to the accumulated probabilities (ACCP), i.e., the

expectation. For better comparison between the observations and reanalysis, COSMO-REA6 is displayed for land grid points, ocean grid points and all grid points.

As displayed in Fig. 3a, of the four mentioned features CS shows the highest proportion with 21.5 % as the most probable feature, followed by the WJ with just under 15 %. The CJ reaches merely 4 %, while the least common feature is CFC with under 1 %. However, NF has a proportion of almost 60 %, thus almost three times as much as the CS. This might be caused

by NaN values within the data set, which are replaced by the mean values of the variable. This complicates the distinction between the features as fewer parameters include information about the current conditions, leading to higher probabilities for NF and causing NF to be the most probable feature more often. Indeed, Fig. 3b shows that the proportion of NF reduces by over 10 percentage points if probabilities for all features and not only the most probable one are taken into account. Although this leads to an increase of all mesoscale features, it is not to the same amount. While the features with overall higher probabilities,

namely WJ and CS as shown in Fig. 1, increase by around 10-30 %, CJ shows an increase of 73 %. The CFC, which shows the

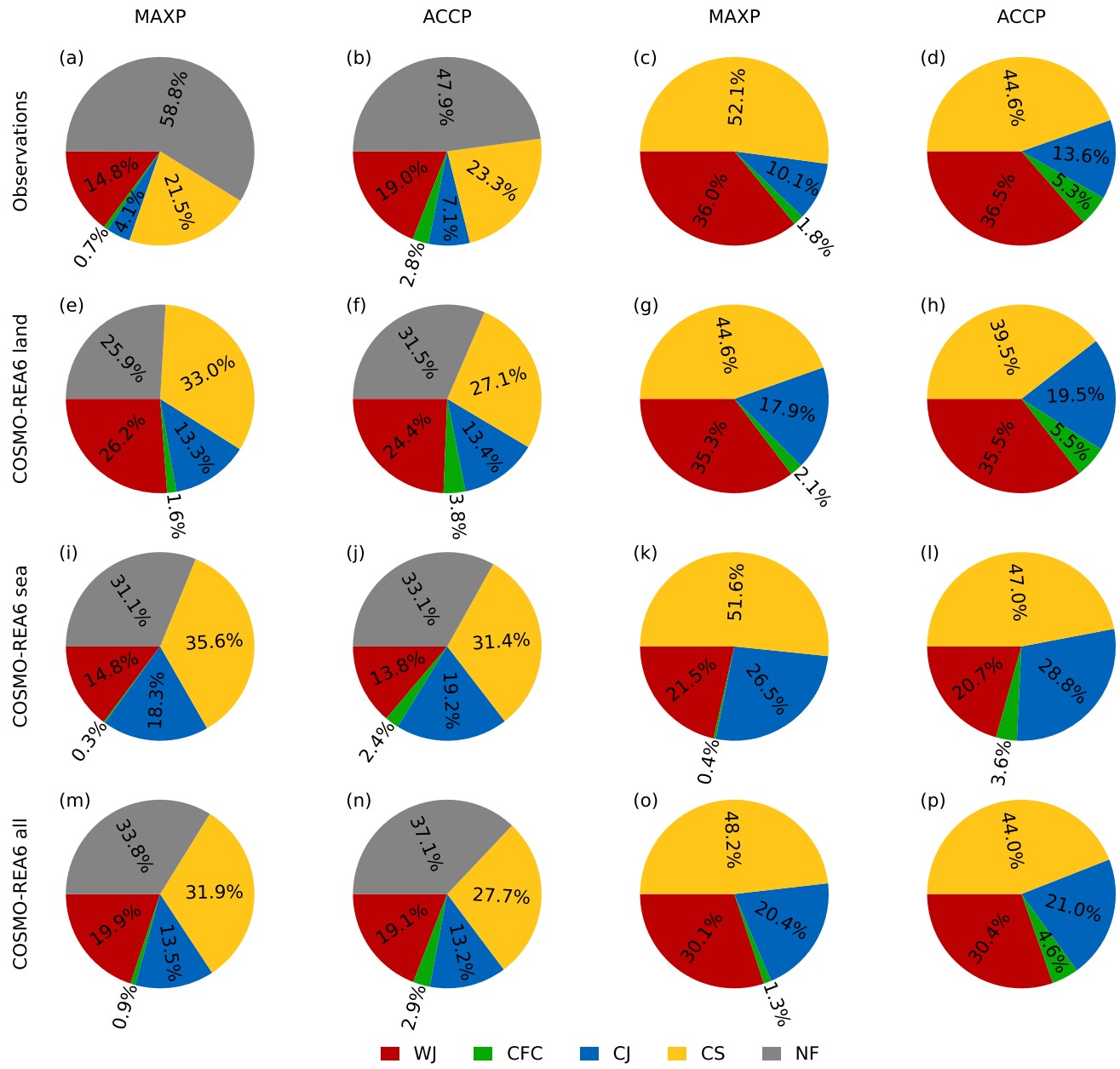

**Figure 3.** Relative frequency of features for station observations (a–d), COSMO-REA6 land grid points (e–h), COSMO-REA6 ocean grid points (i–l) and all COSMO-REA6 grid points (m–p) for the most probable feature (MAXP; first and third column) and accumulated probabilities (ACCP; second and fourth column). The first two columns include NF, while it is neglected in the latter two due to its high frequency.

highest uncertainty, increases by 400 % demonstrating the gain by using the assigned probabilities. Nevertheless, CFC is by far the least common cause for high winds.

Neglecting NF in Fig. 3c,d draws the focus on the ratio of the mesoscale features themselves. In the MAXP perspective, CS and WJ are the cause of high winds in over 50 % and 36 %, respectively, while the more damaging features CJ and CFC (e.g.
Hewson and Neu, 2015; Earl et al., 2017) only show a proportion of 10.1 % and 1.8 %, respectively. However, these are also the features with lower certainty (Fig. 1). Hence, in the ACCP perspective, CJ and CFC come to a total of around 19 %, i.e., an increase of around 60 %. While the WJ also increases slightly, the proportion of CS decreases by almost 8 percentage points. This suggests that the CJ and CFC mostly lose against the CS to be the most probable feature.

In contrast to the station observations, the proportion of NF is considerably lower in COSMO-REA6 data as seen in Fig. 3e,f.
This supports the hypothesis that the high proportion in observations is due to NaN values, as the data are of course complete for all parameters here. Still, the proportion of NF is accountable for around one quarter to one third of high winds. This is due to several reasons discussed below and in Sect. 3.3 as well as overall higher uncertainty in uncommon cyclone developments, such as double fronts. The reader is referred to Sect. 7 of Part 1 for a detailed discussion (Eisenstein et al., 2022a). As the overall certainty of NF is lower for COSMO-REA6 and closer to the density of CS and WJ (solid lines in Fig. 1), the proportion in
Fig. 3f actually increases, while the CS and WJ proportion decreases. As the proportion of NF substantially affects also the proportions of the other features, we compare observations and gridded data without NF from here on (Fig. 3g,h). So apart from that, the largest difference between the two data sets can be seen in the percentage of the CJ with an increase of almost 8 percentage points for land grid points in COSMO-REA6 and over 16 percentage points for ocean grid points for MAXP. This is due to the fact that the CJ commonly occurs in northern continental Europe, over the sea and in Scandinavia, i.e.,
regions where fewer station observations are available in our data set (see Sect. 3.2 and Fig. 6e). On the other hand, CS mostly occurs further south, i.e., farther away from the cyclone centre, such that a higher percentage in CJ leads to lower percentages for CS in the gridded data over land. The WJ, which occurs more over land than over sea (comparing Fig. 3e–h and i–l), shows similar percentages on both observations and COSMO-REA6 over land. With similar probability distributions for the mesoscale features for both data sets (Fig. 1) it is again apparent that CFC shows a higher percentage in ACCP. Comparing
sea and land grid points (Fig. 3e–h vs. i–l) finally shows that CJ and CS occur more often and in a wider area over the ocean compared to CFC, which almost exclusively occurs over land, where friction is higher and static stability lower during daytime. As mentioned before, WJ is more common over land with around 15 percentage points more in both MAXP and ACCP.

Finally, Fig. 3m–p show the proportions for all grid points. Note that the number of land grid points is around 35 % higher than the number ocean grid points. Overall, almost half of windy conditions are caused by CS, followed by the WJ and CJ
with around 30 % and 21 %, respectively. Again, for CFC the difference between MAXP and ACCP shows an considerable difference from 1.3 % to almost 5 %. To examine how robust these numbers are, we computed three subsets of nine randomly chosen winter seasons. The proportions vary just slightly with an average of around 2 % (not shown), as to be expected considering the small fluctuations between winter seasons (see discussion of Fig. 5 below).

As the overall frequencies are similar and differences plausible in both data sets and COSMO-REA6 has the advantage of
an homogeneous field without missing parameters, we focus on the gridded data set from here on.

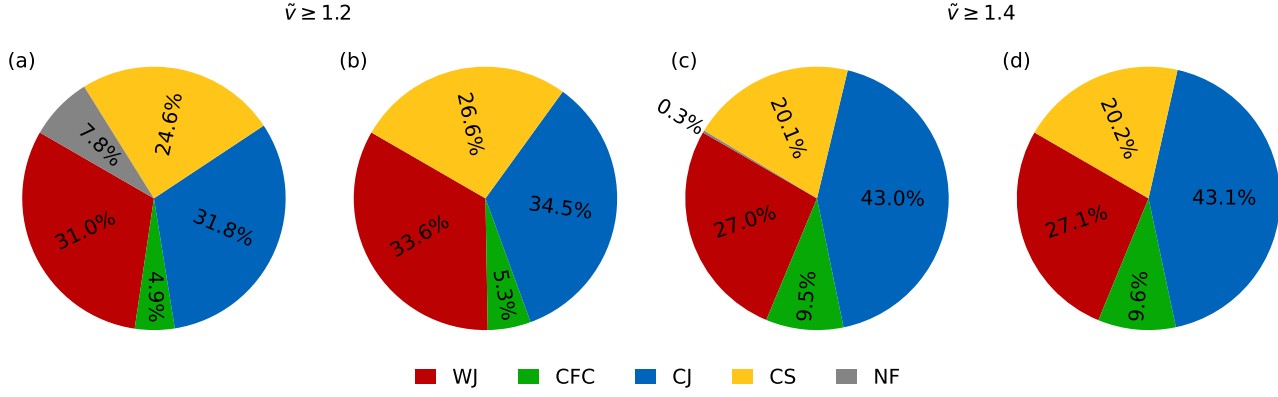

**Figure 4.** As Fig. 3 m,o (COSMO-REA6 all; MAXP) but for grid points with (a),(b) $\tilde{v} \geq 1.2$ and (c),(d) $\tilde{v} \geq 1.4$.

Figure 4 shows the proportions of each feature for MAXP analogously to Fig. 3m,o (COSMO-REA6 all) but only including grid points where $\tilde{v} \geq 1.2$ (left columns) and $\tilde{v} \geq 1.4$ (right columns). Note that this is only the case for around 1.5 % and 0.1 %, respectively, of the previously included data points. Considering only $\tilde{v} \geq 1.2$, the proportion of NF reduces by over 75 % (Fig. 4a), while it decreases to almost 0 % if only grid points with $\tilde{v} \geq 1.4$ are included (Fig. 4c). This suggests that higher

winds within the vicinity of a cyclone are mostly associated with one of the introduced features. When NF is neglected, the proportion of CS is reduced by 45 % to 58 % (Fig. 4b,d) suggesting that while the CS affects a large area, it is less common for it to be the cause of extreme winds. In contrast, the more damaging features CJ and CFC show an increased proportion. CJ shows an increase of about 69 % for $\tilde{v} \geq 1.2$ and 111 % for $\tilde{v} \geq 1.4$. Consistent to Earl et al. (2017), CFC shows an even stronger increase with over 300 % and over 600 %, respectively. These results are consistent with the wind characteristics, as

will be discussed in Sect. 4. Meanwhile, the proportion of WJ decreases by about 10 %, as it usually causes weaker winds compared to the CJ and CFC (e.g., Hewson and Neu, 2015; Earl et al., 2017).

The discussed proportions in Fig. 3 depend not only on the occurrence of the feature but also on its size. Figure 5 shows the seasonal and interannual evolution of stormy time steps and occurrence of each feature, respectively. This is computed as the sum of stormy time steps during which a certain feature is detected as the most probable feature at at least 100 grid points over

the investigation domain, normalised by the number of stormy time steps. This measure is thus independent of the number of grid points, where the feature is identified.

Looking at all stormy time steps of the investigation period (left side of Fig. 5), it is evident that NF (grey dot) occurs somewhere in the domain practically for every moment in time. The same holds for CS (orange dot) with only marginally lower frequencies. Both CJ (blue) and WJ (red) occur in over 80 % of stormy time steps with slightly higher values for CJ.

Together with the lower proportion of CJ compared to WJ as seen in all panels of Fig. 3 except (i)–(l) (over the ocean) this suggests that the CJ is on average a smaller feature than the WJ. The least frequent feature with around 43 % is CFC (green) contributing to the low proportions in all panels of Fig. 3.

With respect to the mean seasonal cycle, the black line in the middle section of Fig. 5 shows that November and March are the least stormy months during our investigation period 2000 to 2019 with a peak in January in between. This is consistent with the cyclone track density plots shown in Fig. B1 of the Appendix B. Somewhat surprisingly, October shows the highest number of stormy time steps of all months. On long-term average, the majority of storms occur between December and February, and thus the 98th percentile of wind speeds is highest for that period and lower towards autumn and spring, consistent with Feser et al. (2015); Laurila et al. (2021). However, the recent two decades show a larger number of noteworthy storms in October (e.g., Christian 2013, Xavier 2017, Herwart 2017) compared to November, which leads to a larger number of stormy time steps in the 19 years. This difference might be further enhanced by the slightly lower 98th percentile of $v$ in October compared to November. The higher frequency is consistent with October showing a slightly higher cyclone density than November (Fig, B1a,b). With respect to the individual features, the rather rare CFC has a marked seasonal cycle with an apparent peak in December and January. The WJ has a smaller relative peak during peak winter months. While the relative frequency of CJ is only slightly higher than for WJ, its frequency increases with the winter passing leading to a maximum of over 90 % in March. Recall that the $\tilde{\theta}$ predictor is a normalised parameter, such that a cooling Arctic with progressing winter and possibly more cold-air outbreaks, i.e., lower values of $\tilde{\theta}$, might lead to higher numbers of high winds being allocated to the CJ – and CS. Both NF and CS are so frequent that an annual cycle is not evident in this analysis.

Finally, the right section of Fig. 5 shows the interannual evolution of stormy time steps and wind features. Overall, lower numbers of stormy time steps, such as 2002/03, 2005/06, 2009/10, 2010/11 and 2012/2013 are consistent with negative values of the North Atlantic Oscillation (NAO; e.g., Wanner et al., 2001), which describes the large-scale circulation over the North Atlantic and originally represents the pressure difference between Iceland and the Azores (Hurrell, 1995). For our study area, slightly positive NAO values facilitate stormy conditions, as the typical cyclone paths for such NAO conditions correspond to tracks over the British Isles, the North and the Baltic Sea. Accordingly, most peaks are associated with positive NAO phases. Furthermore, quieter winter seasons consistent with literature such as 2010/2011 (e.g., Santos et al., 2013; Laurila et al., 2021) can be found. With 2009/10 being a particular cold winter season (Wang et al., 2010), the occurrence of detected WJs is lower compared to other winters, while CJ shows a peak. The peaks of WJ and CFC in 2015/16 are consistent with the winter season being particularly wet and warm as discussed in McCarthy et al. (2016). Again, NF and CS occur too frequently to detect an interannual cycle. Overall, all features have no to very weak positive correlation with the number of stormy time steps ($0 - 10 \%$). The coefficient of variance is lowest for CS and stormy time steps with 41 % and highest for CFC with 54 %.

With respect to long-term trends, a slight decline is evident consistent with the overall decrease in the number of winter storms in a warming climate (Catto et al., 2019). However, given that our investigation period covers only 19 years, a Mann-Kendall test (significance level of 0.05; Hussain and Mahmud, 2019) did not indicate statistical significance in any of the time series.

## 3.2 Earth-relative statistics

An Earth-relative framework enables us to learn which regions are commonly affected by which feature. Figure 6 shows a geographic distribution of the relative frequency of NF and the four mesoscale wind features (MAXP) as well as of overall

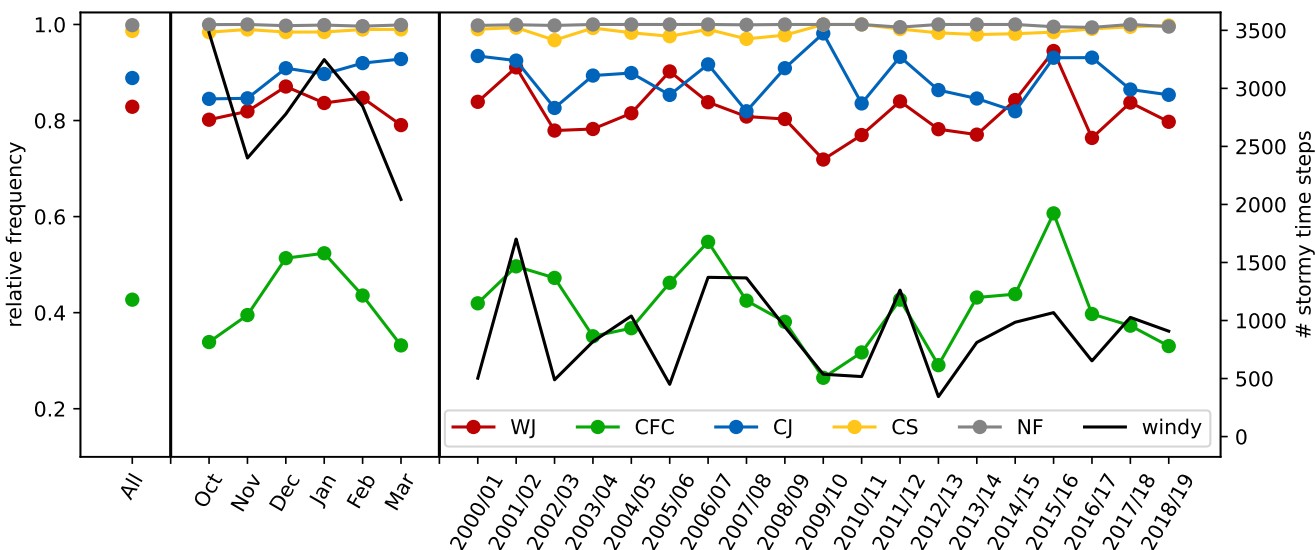

**Figure 5.** Number of stormy time steps (black line and right axis) and number of time steps with the feature occurring divided by the number of stormy time steps (left axis) for all time steps (left), each month (middle) and each winter season (right).

windy conditions. An analogous plot for ACCP can be found in the Appendix B showing overall similar properties (Fig. B2). Results are displayed on a $0.5° \times 0.5°$, thus aggregating over 16 grid points.

The number of windy conditions, i.e., $\tilde{v}$ exceeding 0.8, in proximity to the cyclone centre (within $\pm 15°$ in zonal direction and $-15°$ to $5°$ in meridional direction) is displayed in Fig. 6a. This criterion leads to the highest numbers over the North Sea, Denmark, northern Germany and the Baltic Sea, i.e., south of the maximum track density shown in Fig. 2. Furthermore, choosing a threshold of 80 % of the 98th percentile results in more windy conditions over the ocean (see Appendix A; Fig. A1), hence, the absolute frequencies of each feature are normalised by the number of exceedences for each grid point. Due to orographic effects and higher noise in these regions, we exclude grid points above 800 m (hatching in Fig. 6) and the area east of the Alps, including Hungary, Slovenia and the Balkans. Note that the frequencies of NF, WJ, CFC, CJ and CS add up to 1.

Figure 6b shows that NF is most common with a relative frequency of around 50 % in the periphery of the area, i.e., north and further south of the most common cyclone paths and wind footprints. Although removing areas of high orography itself, their effects can still be seen upstream, when the mostly westerly winds encounter mountain barriers such as the Scandinavian mountains, Western Alps and Carpathians leading to higher frequencies of NF in these regions. Over Western and Central Europe, high winds are usually closer to the cyclone centre, such that they are mostly associated with one of the mesoscale features.

The WJ occurs mostly over Western Europe with relative frequencies of almost 40 % and decreases over Germany (Fig. 6c) down to 20 % over Poland. This is consistent with the occurrence early in the life cycle when the cyclone is still in the western regions. The higher frequency east of the Scandinavian mountains should be treated with caution as it might be caused by

orography such as the foehn effect when a cyclone crosses the mountains. Moreover, a land-sea contrast is visible, which is weaker in ACCP (Fig. B2). A possible explanation are the different thermal characteristics of land and ocean, such as daytime heating leading to differences in $\tilde{\theta}$. Hence, just trained over land, RAMEFI might have difficulties distinguishing the WJ from the CS over sea in ambiguous situations. However, looking at exemplary winter storm cases, where features are well developed, a difference in the detection over ocean and land grid points is not evident (Eisenstein et al., 2023b).

CFC shows low frequencies under 4 % for MAXP (Fig. 6d), while values are twice as high for ACCP (Fig. B2d) due to the lower certainty of the feature (Fig. 1) as discussed in Sect. 3.1. However, a distinct land-sea contrast is visible in both, where CFC seems to be almost exclusively detected over land. This might be due to land effects, such as frictional convergence and land surfaces being heated up more strongly than over ocean during the day, leading to destabilisation of the atmosphere. Moreover, CFC develops slightly later than the WJ when the cold front intensifies (Sect. 3.3 and Fig. 7b). Note the patchy

behaviour over land, possibly caused by local small-scale effects due to, among other things, surface roughness and orography. Distinct maxima are found east of or over mountainous regions, such as the Scottish Highlands and the Scandinavian mountains. Here, again, results given by RAMEFI should be treated with caution. As orography can induce convection, CFC might be detected without the occurrence of a cold front, but where high wind speeds are associated with a strong pressure gradient or other features combined with orographic convection.

As expected, the occurrence of the CJ (Fig. 6e) maximises in the northern half of the domain, much farther north than for WJ (Fig. 6c), with a distinct footprint over the northern British Isles, the North Sea and the Baltic Sea. Over the British Isles, where intense cyclones are more frequent than over the North Sea (not shown), the CJ shows a maximum with over 20 %. Overall, the CJ occurs mainly over the sea and coastal areas (although RAMEFI was trained over land).

   CS shows high values in the Bay of Biscay, where other features are rarely detected, and a rather abrupt drop over France in

the east (Fig. 6f). Together with the opposite patterns for WJ this suggests a possible false detection in some cases. As explained above, we suspect some systematically different behaviour between land and ocean to be at least partly responsible for this. A second peak in CS can be found over Eastern Europe, where most other features have already weakened at that late stage in the cyclone life cycle. Overall, the CS occurs further south than the CJ and thus farther away from the cyclone centre.

### 3.3    System-relative statistics

For the system-relative framework, we concentrate on the area within $\pm15°$ in zonal direction, $-15°$ and $+5°$ in meridional direction of the cyclone centre. This translates to around $\pm1073\,\mathrm{km}$ in zonal direction at $50°$ latitude and $1670\,\mathrm{km}$ and $557\,\mathrm{km}$ in southern and northern direction, respectively. Figure 7 shows a composite over the 19 extended winter seasons from 2000–2019 relative to the cyclone centre (a) and life cycle (b).

   With respect to the mean spatial distribution, the WJ mostly occurs within the southeastern quadrant of a cyclone consistent

with conceptual models (see Fig. 1 in Part 1; Eisenstein et al., 2022a). As shown in Fig. 7a, the WJ usually has a distance from $250\,\mathrm{km}$ to $1500\,\mathrm{km}$ from the centre. The CFC occurs around $3°-5°$ farther to the west, i.e., upstream with respect to a westerly flow and also slightly shifted to the north, closer to the cyclone centre. Since CFC is a relatively small elongated and narrow feature (as is the front itself), the location is harder to pinpoint over so many cases and the location varies the most from case

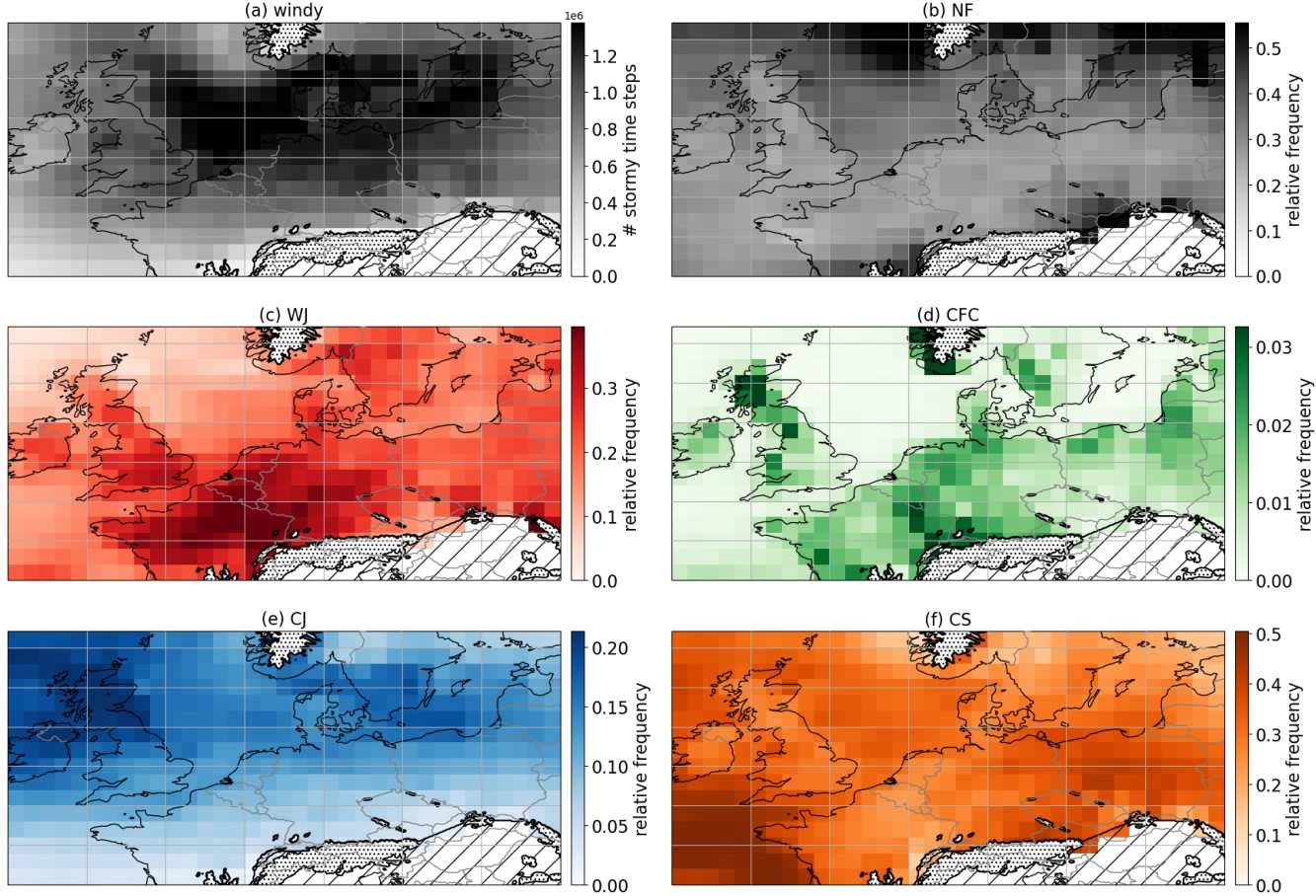

**Figure 6.** Relative frequency of (a) windy conditions, (b) NF, (c) WJ, (d) CFC, (e) CJ and (f) CS. (b)–(f) are normalised by (a). Results are displayed on $0.5° \times 0.5°$ boxes. Hatching indicates grid points with an altitude above $800\,\mathrm{m}$ (dots) and excluded areas east of the Alps (lines).

to case compared to the other features. Thus, although CFC overlaps with other features statistically, this should usually not be the case for individual cyclones. The CJ is situated to the southwest to south of the cyclone centre with some statistical overlap with CFC. It occurs closer to the centre than the CS, which dominates the southwest quadrant and can have a distance of up to $1500\,\mathrm{km}$ from the cyclone centre. The $25\,\%$ shaded area for NF is split into two patches. The majority of NF is detected to the southeast to south of the cyclone centre, mostly coinciding with the WJ but extending its reach to the north and southwest. The northern part of this patch is located in the area of the warm front and is possibly connected with the CCBa as discussed in Earl et al. (2017) and Gentile and Gray (2023). A second, smaller patch can be found to the northwest of the cyclone centre. In this region, the CJ usually occurs before it is wrapped around the cyclone centre. As the CJ follows the bending of the front, the wind direction differs from the CJ later on, when it wraps around the centre, such that RAMEFI does not identify this part

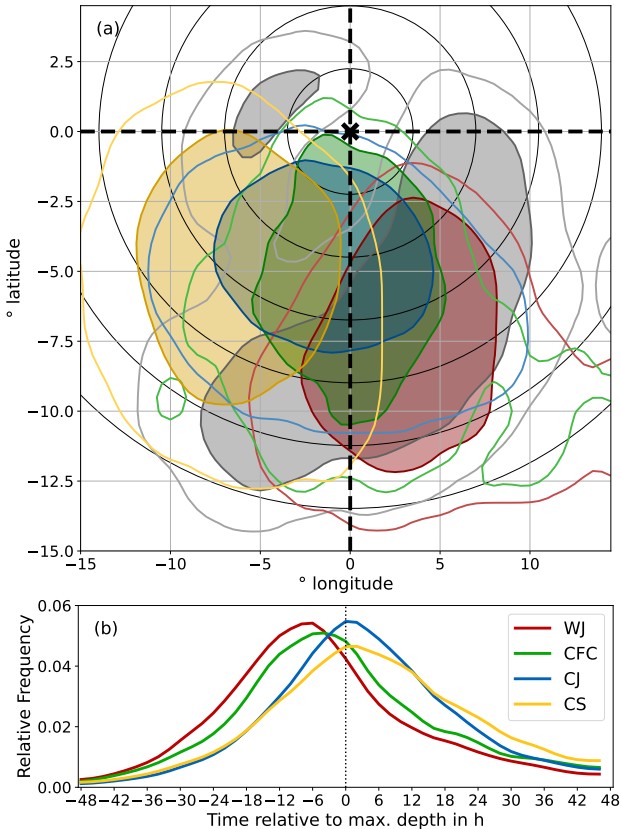

**Figure 7.** Occurrence of the identified mesoscale wind features (a) relative to the cyclone centre –including NF in grey– and (b) relative to the cyclone life cycle. The contours in (a) show the area with most feature occurrences, including 25 % (filled contours) and 50 % (outer contours) of the detected features. Black circles show the distance to the cyclone centre in 250 km increments using 50 °N as a reference latitude.

of the CJ. However, high wind speeds in this area are usually only caused for very strong CJs, as the relative movement of the air in this area is against the cyclone motion, weakening the Earth-relative wind speed, particularly in fast-moving weaker
cyclones (Eisenstein et al., 2020). As shown in Appendix B, the relative frequency of the features shows that the NF mostly occurs in the north-eastern quadrant, where other features are rare, and where windy conditions are less common (Fig. B3).

Figure 7b shows the relative frequency of the four mesoscale wind features throughout the lifetime of the parent cyclone from 2 days before until 2 days after the time of maximum depth, i.e., the deepest pressure minimum during a cyclone's life cycle (marked as 0 in Fig. 7b). The WJ is the first feature to develop with a maximum at $-6$ h. 50 % of detected WJ grid points occur
between $-18$ h and the time of maximum depth. With a small offset of around $2-3$ h the CFC follows, consistent with its more western location in Fig. 7a. The peak is slightly lower at around $-4$ h and 50 % of CFC points are detected between $-16$ h and 6 h. Contrary to Hewson and Neu (2015), the CJ develops already several hours before the time of maximum depth. However,

the peaks of CJ and also CS are around 0, even though the peak of CS is lower. The WJ and CFC are already decreasing at that time, as the warm air begins to occlude consistent with Hewson and Neu (2015). While the CJ occurrence decreases faster, the CS stays on for longer. Overall, 50 % of CJ and CS occur within $-8\,\mathrm{h}$ and $10\,\mathrm{h}$ and $-8\,\mathrm{h}$ and $14\,\mathrm{h}$, respectively.

When combining both composites, we can see how the features occur in different locations during the cyclone's life cycle. This is displayed for four exemplary time slots in Fig. 8. An animation showing all time slots from $24\,\mathrm{h}$ before to 24 after time can be accessed in the Supplement (Eisenstein et al., 2023b). $24\,\mathrm{h}$ before the time of maximum depth, only WJ and CFC appear in the composite, the round maximum of WJ to the south-southeast of the cyclone centre and the CFC with a more north–south elongated maximum closer to the cyclone centre (Fig. 8a). CJ and CS develop in the following hours to the southwest of the cyclone centre, while the size of the WJ increases as shown $12\,\mathrm{h}$ later (Fig. 8b). The area of CFC also increases, however the increase in width is probably rather due to the variation in location than an increase in size. Around the time of maximum depth, as shown in Fig. 8c, the WJ area decreases and shifts farther away from the centre in southeastern direction, now with a stronger west–east orientation. The CJ has increased in size and now stretches across both southern quadrants, whereas the CS fills most of the southeastern quadrant. Furthermore, NF covers most of the area overlapping with all other features. As mentioned above, the area northeast to north of the cyclone centre, which does not overlap with any of the mesoscale features, corresponds to the CCBa as described in Earl et al. (2017) and Gentile and Gray (2023). $12\,\mathrm{h}$ after time of maximum depth (Fig. 8d), WJ and CFC have mostly vanished, while CJ and CS are much diminished in size.

Overall, these results are mostly consistent with idealised schematics and conceptual models in the literature (e.g., Hewson and Neu, 2015, their Fig. 1). However, the very large set of differing cyclone developments in our comprehensive data set is able to show a larger variety. Since our study domain is too small to cover the whole development of the investigated cyclones, especially the early stages of a feature might be missed, such that an analysis of feature duration and comparison with the literature (e.g., Hewson and Neu, 2015) is not meaningful. Over the investigation domain, the WJ, CJ and CS have a broadly similar duration, with an average of around $20\,\mathrm{h}$ (not shown). In contrast, the smaller and rarer CFC appears only half as long with an average of around $11\,\mathrm{h}$ (not shown).

## 4   Characteristics of high-wind features

Using RAMEFI over a 19-year time period also allows us to analyse the distributions of selected meteorological parameters for each feature, i.e., finding characterisations of meteorological conditions. By construction, the eight parameters used for the training of RAMEFI (Sect. 2.3) behave as already documented in Part 1 and are therefore not displayed here, but only briefly discussed. Instead we concentrate on wind speed ($v$) and gusts ($v_{\mathrm{gust}}$), the gust factor ($g_{\mathrm{v}}$), specific and relative humidity ($q$ and $RH$, respectively) and total cloud cover ($cc$). Figure 9 shows boxplots for these parameters for all grid points, while a distinction between land and sea grid points can be found in Appendix B (Fig. B4).

With respect to $p$ (not shown), the CJ has the deepest pressure being closest to the cyclone centre, while WJ and CS show higher values. In contrast to Fig. 10 in Part 1 (Eisenstein et al., 2022a), the CS has no second peak at low $p$, which was due to the exceptionally deep storm Sabine (February 2020) included in the training. This is not the case here, and even if it was

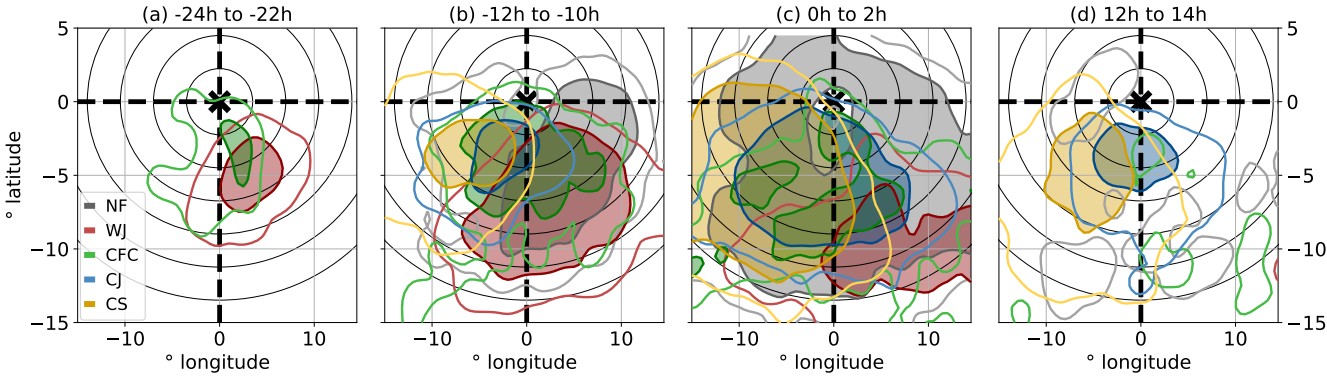

**Figure 8.** As Fig. 7a but for different times relative to the cyclone life cycle, i.e., time of maximum depth: (a) -24 h to -22 h, (b) -12 h to -10 h, (c) 0 h to 2 h and (d) 12 h to 14 h.

included, it would carry much less weight in a composite of almost 20 years. Being usually ahead of the cold front, WJ shows falling pressure, while pressure rises in the CS and CJ areas. Furthermore, WJ and CFC show warmer $\tilde{\theta}$ compared to CJ and CS (not shown). $RR$ values over $1\,\mathrm{mm\,h^{-1}}$ are only common for CFC. CFC is characterised by slightly positive $\Delta d$ values, while hardly any wind shift is found for WJ, CJ and CS.

As seen in Fig. 9a, $v$ is usually highest for the CJ with the median being around $15\,\mathrm{m\,s^{-1}}$ and the 99th percentile over $25\,\mathrm{m\,s^{-1}}$, making it the most common cause for high winds (e.g., Hewson and Neu, 2015; Gentile and Gray, 2023). While CS and WJ show similar 99th percentiles at around $23\,\mathrm{m\,s^{-1}}$, the median of WJ at around $10\,\mathrm{m\,s^{-1}}$ is about $3\,\mathrm{m\,s^{-1}}$ lower than the median of CS. NF has a slightly higher median than CS, but a similar mean value. The lowest values are found for CFC with a median of under $10\,\mathrm{m\,s^{-1}}$. Naturally, $v$ over the ocean is considerably higher than over land (Fig. B4) affecting the overall $v$
depending on how often the features occur over land or ocean (Fig. 3e–l and Fig. 6). $v_{\mathrm{gust}}$ shows similar behaviour for CJ, CS, WJ and NF. However, CFC shows the second highest gusts with up to $35\,\mathrm{m\,s^{-1}}$ (Fig. 9b). This leads to the highest $g_v$, which simply displays the ratio between $v_{\mathrm{gust}}$ and $v$ and reaches 3 in case of CFC (Fig. 9c). This is not surprising, as convection is associated with high instability and turbulence. Although both wind and gust speeds are much higher over the ocean, the gust factor differs significantly between land and sea (Born et al., 2012) with less friction and other causes of turbulence over the
sea leading to a weaker increase of gust speeds compared to wind speeds (Fig. B4a–c).

With respect to moisture and cloud variables, CFC shows the highest values of specific and relative humidity, followed by the WJ in the warm sector, CJ and lastly CS (Fig. 9d,e). CS may also include high winds caused by dry intrusions (Raveh-Rubin and Catto, 2019; Catto and Raveh-Rubin, 2019), leading to overall drier conditions for this feature. Moreover, especially the SJ, which is here included in the CJ feature, occurs in the dry slot area. Following the Clausius-Clapeyron relation, it is
also intuitive that warmer temperatures enable higher values of $q$. Consistently, all features have lower $q$ values and higher $RH$ values over land. Figure 9f shows the total cloud cover. While stratocumulus and stratus clouds are common in the warm sector ahead of the cold front, cloudless areas can still be found for the WJ in contrast to CFC. Both CJ and CS show a wide

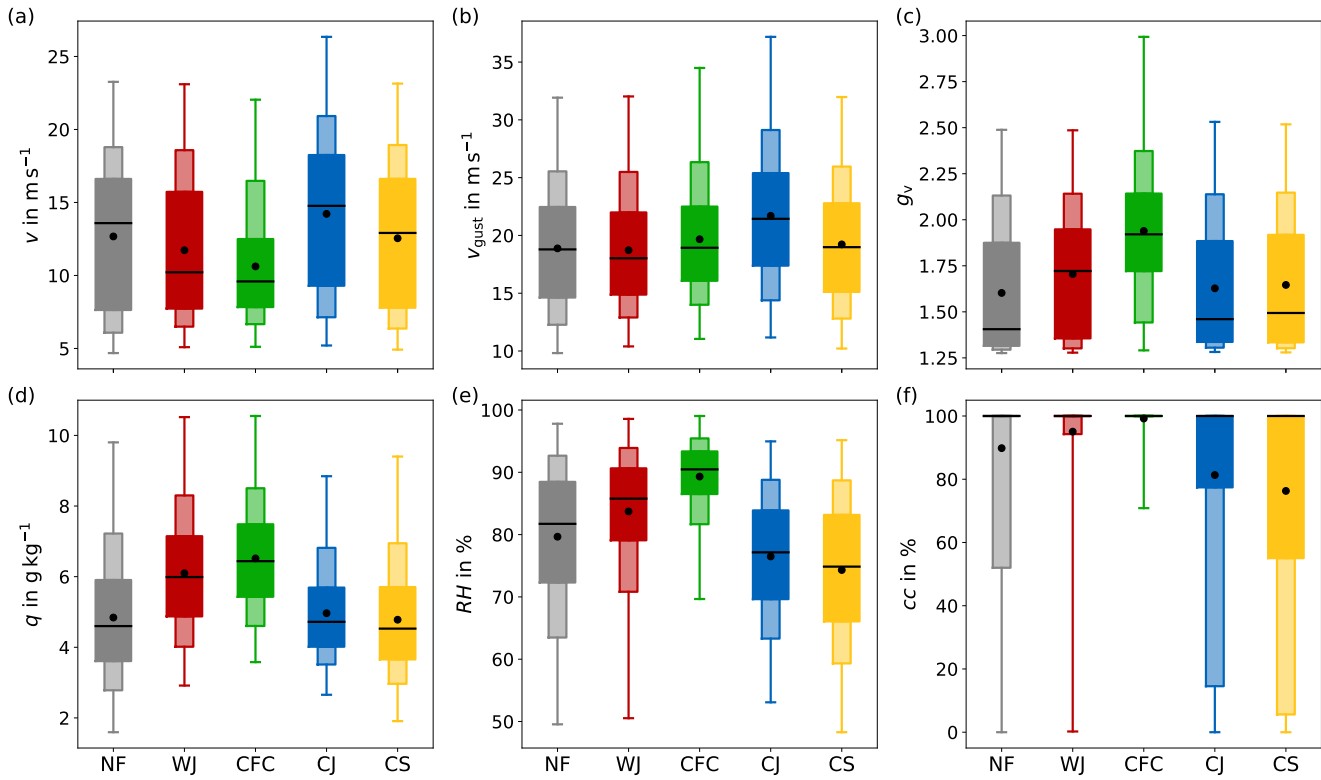

**Figure 9.** Boxplots of each feature for (a) $v$, (b) $v_{\mathrm{gust}}$, (c) $g_{\mathrm{v}}$, (d) $q$, (e) $RH$ and (f) $cc$. Broader boxes show the 25th to 75th percentile, thinner ones the 10th to 90th percentile and whiskers the 1st and 99th percentile. Lines and dots indicate the median and the mean, respectively.

distribution. While the CJ appears at the tip of the cloud head, partly below and slightly ahead of it, allowing for both cloudy and cloudless conditions, the CS is often associated with post-CFC, and thus mixture between cloudless skies and showers. At

times, NF is detected along the warm front (Fig. 7), which is characterised by cloudy conditions, but can also show lower $cc$ values in other areas, e.g., in the warm sector.

## 5  Conclusions

Damaging winds accompanying extratropical cyclones can be caused by several mesoscale features with different characteristics and, thus, also differing forecast errors and potential for damage. To analyse these differences, we developed a novel

objective and flexible probabilistic identification tool called RAMEFI (RAndom-forest based MEsoscale wind Feature Identification) recently introduced in Part 1 of this work (Eisenstein et al., 2022a). The method is trained on the basis of surface observations for 12 storm cases but due to spatial independence and removing location-specific effects, once trained, it can be applied to gridded data without any modification. Here RAMEFI is used to compile a – to the best of our knowledge – first-ever long-term objective climatology of the four wind features WJ, CJ, CS and CFC based on station observations and a

high-resolution reanalysis data set (COSMO-REA6) for a time period of 19 extended winter seasons, i.e., October to March. Using the reanalysis data also allows investigating ocean grid points. Although a systematic validation as done for land in Part 1 is not performed for ocean areas, a subjective inspection of several cases during the analysed time period did not reveal fundamental differences. However, due to a different shape of the wind speed distribution, the threshold of $\tilde{v} > 0.8$ used to define windy conditions results in more frequent occurrence over the ocean than over land.

The considered area includes Western and Central Europe, however, excluding grid points above $800\,\mathrm{m}$ altitude and the Balkans. Focusing on grid points with windy conditions within the vicinity of a cyclone centre, i.e., $\pm 15°$ in zonal and $-15°$ and $+5°$ in meridional direction, we determined the relative frequency of mesoscale wind features in both an Earth-relative and cyclone-relative framework. Furthermore, distinctive characteristics in wind (wind speed $v$, gust speed $v_{\mathrm{gust}}$, gust factor $g_{\mathrm{v}}$ and humidity parameters (specific humidity $q$, relative humidity $RH$, cloud cover $cc$) were investigated. The main findings of

the climatological analysis for the individual features are:

*Warm jet (WJ)*

- Characterised by decreasing $p$, warm temperatures, almost no precipitation and mostly south-westerly winds

- First to occur within the southeastern quadrant of a cyclone, peak around $6\,\mathrm{h}$ before the time of maximum depth

- Detected mostly over land in more than $80\,\%$ of stormy time steps

- Most common over southern UK, France, Benelux states and Germany

- Mostly cloudy conditions; warm sector allows for rather humid conditions

*Cold-frontal convection (CFC)*

- Associated with heavy precipitation, a shift in wind direction and cooling temperatures

- Narrow feature along the cold front, least common, location varies considerably from case to case

- Occurs almost exclusively over land where, e.g., daytime heating and frictional convergence, can strongly enhance the development of convection along the cold front, and is also detected particularly around mountainous areas (sometimes orographic triggering independent of cold front)

- Highest $g_{\mathrm{v}}$ with rather low $v$ and high $v_{\mathrm{gust}}$ up to $35\,\mathrm{m\,s^{-1}}$ and higher in extreme cases

- Highest values of $q$, $RH$ and $cc$ connected with the convection

*Cold jet (CJ)*

- Shows increasing but overall deepest $p$, westerly winds and cold temperatures

- More frequent over sea than land in over $80\,\%$ of stormy time steps mostly affecting northern UK, the North and Baltic Seas, Scandinavia and northern Germany

- Smaller feature than WJ, occurring close to the cyclone centre, first to the southwest and later to the south of it

- Usually cause of highest winds and gusts

- Cloudy conditions below cloud head and drier at the tip of it in dry-slot region

*Cold sector (CS)*

- Associated with cold temperatures, westerly winds, increasing and higher $p$ compared to CJ
- Occurs in almost all winter storms and time steps affecting a relatively large area in the southwestern quadrant of a cyclone
- Last to decay, hence most common cause of high winds over eastern Europe
- Sunnier conditions with patches of post-cold frontal convection
- Overall drier conditions due to dry intrusion

The locations of the features relative to the cyclone centre found in this climatology are mostly consistent with conceptual models based on case studies or subjective identification (see Fig. 1 of Eisenstein et al., 2022a). While previous literature suggests a cyclone to be first dominated by the WJ and second by the CJ (e.g., Hewson and Neu, 2015; Rivière et al., 2015), this climatology further revealed the occurrence of CFC in early development stages and the dominance of CS in later ones. Also further characteristics in wind and humidity parameters show mostly consistent behaviour to previous studies (e.g., Hewson and Neu, 2015; Earl et al., 2017). The large number of storms investigated helped to reveal the large variability in the location of CFC in a system-relative framework, similar to the blurring of frontal boundaries in composites discussed in Dacre et al. (2012). Other differences to the literature include the time of occurrence of the CJ already several hours before the time of maximum depth in contrast to Hewson and Neu (2015). Overall, RAMEFI allows for a more objective and more thorough analysis and description of the mesoscale wind features. This climatology demonstrates the applicability of RAMEFI for longer time periods and data it was not trained on. The new data set can serve the community as a climatological reference for case studies or in combination with other objective climatologies (e.g., Sprenger et al., 2017). In future work we plan to use this climatology for a feature specific forecast error analysis and to explore the potential for feature-dependent post-processing. This will ultimately show whether the differences in stability, turbulence and shallow and deep convection between the features do in fact lead to different physical error characteristics that can be corrected statistically in a more targeted way, helping to improve wind and gust forecasts and warnings.

*Code availability.* RAMEFI is available at https://gitlab.physik.uni-muenchen.de/Lea.Eisenstein/ramefi, where it will be updated in future studies, and is archived at https://doi.org/10.5281/zenodo.6541303 (Eisenstein et al., 2022b) at the time of submission of Part 1 (Eisenstein et al., 2022a).

*Data availability.* COSMO-REA6 data are available under https://reanalysis.meteo.uni-bonn.de (Hans-Ertel-Centre for Weather Research, 2019). The observation data over Europe was provided by DWD for this work and cannot be made freely available. The reader is advised to

 contact the DWD directly regarding these data (klima.vertrieb@dwd.de). Values of the NAO phases are provided by the Climate Prediction Center at the National Oceanic Atmospheric Administration (NOAA). The output of RAMEFI as well as files to filter the output as described in Sect. 2.4 are available at https://doi.org/10.5281/zenodo.8370478 (Eisenstein et al., 2023a).

*Video supplement.* The video supplement showing exemplary winter storms occurring within our studied time period and an animation of system-relative occurrence over time relative to the cyclone life cycle can be freely accessed at https://doi.org/10.5281/zenodo.7729357
Eisenstein et al. (2023b).

## Appendix A: Wind distribution

In Part 1 of this work, a threshold of 80 % of the 98th percentile of high wind speeds was introduced to define windy conditions. While the focus of Part 1 was on stations / grid points over land, we also include ocean grid points for the climatology. However, due to varying friction, orography, heating of the surface and more, the wind distribution over the ocean is has a fundamentally
different form compared to land (e.g., Wieringa, 1973; Born et al., 2012), as displayed for exemplary locations in Fig. A1. The right tail of the distribution shows considerably stronger winds leading to a higher number of time steps exceeding 80 % of the 98th percentile. Overall, the threshold is exceeded around 45 % more often over the ocean compared to land. To nevertheless allow a fair comparison, we normalise the occurrence by the number of time steps with windy conditions.

## Appendix B: Further figures

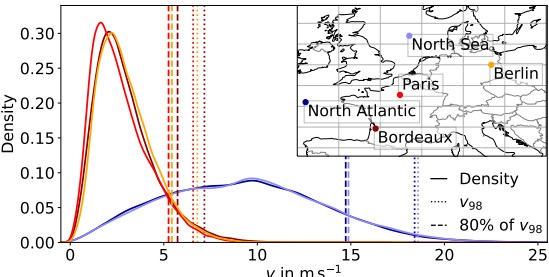

**Figure A1.** Distribution of $v$ for two ocean grid points in the North Atlantic (dark blue) and the North Sea (light blue) and three land grid points close to Bordeaux (dark red), Paris (red) and Berlin (orange), respectively. Dotted lines mark the 98th percentile, dashed lines 80 % of the 98th percentile.

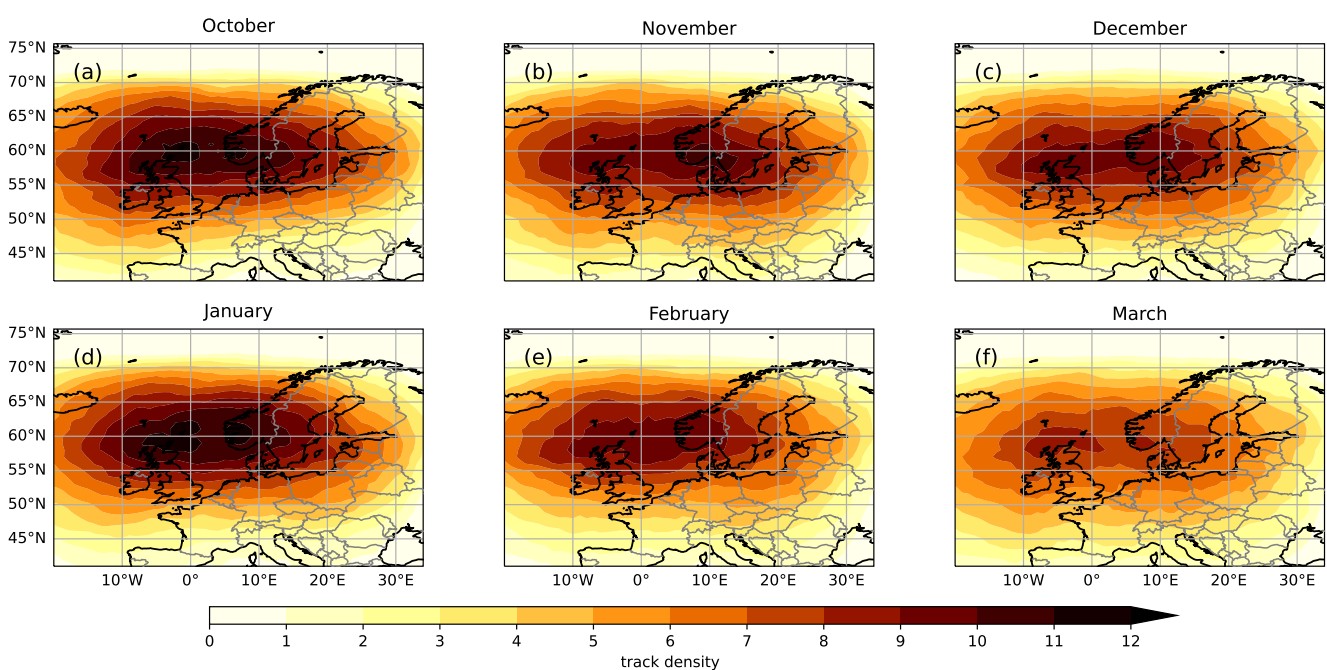

**Figure B1.** As Fig. 2 but for each month considered in the climatology.

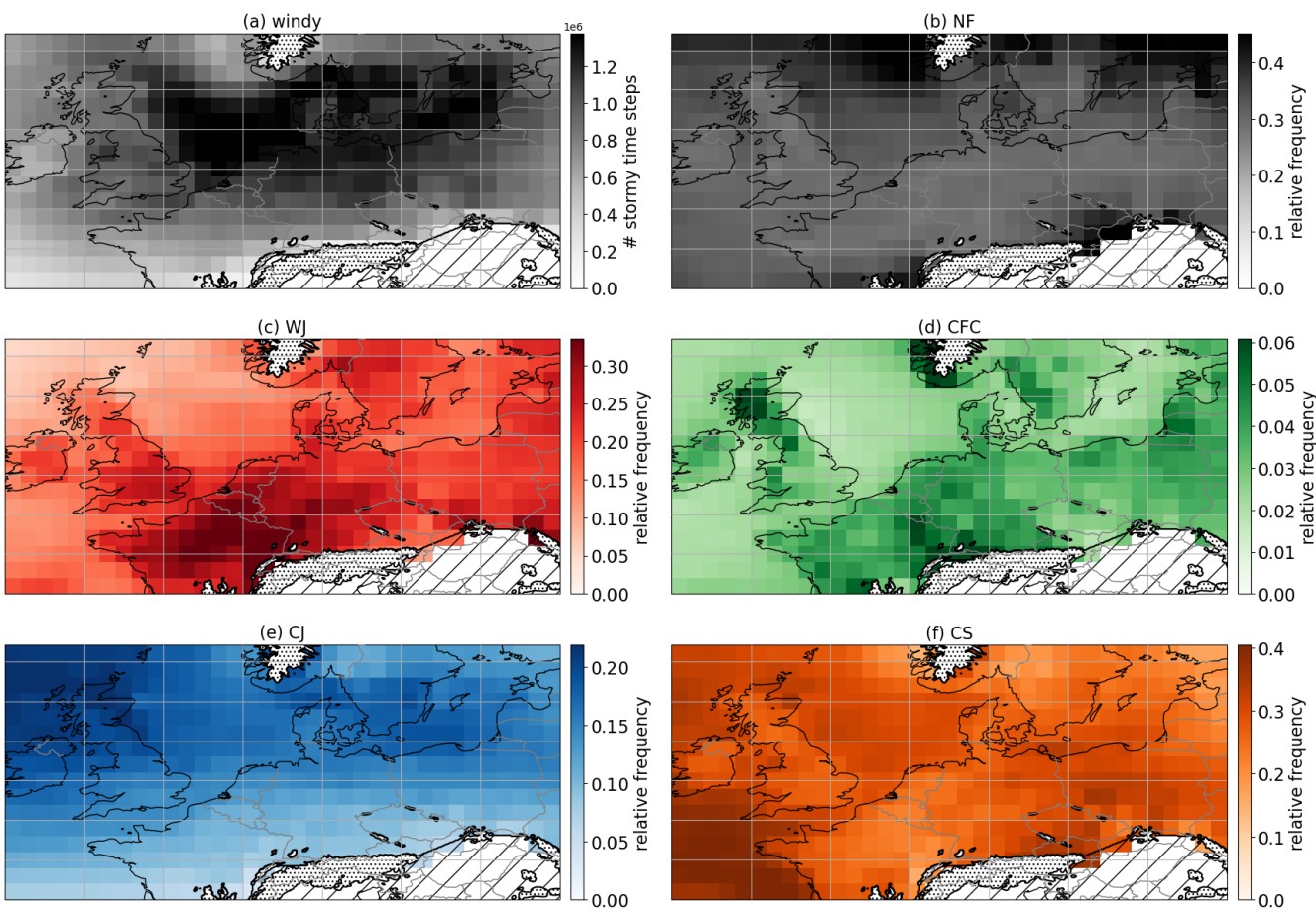

**Figure B2.** As Fig. 6 but ACCP.

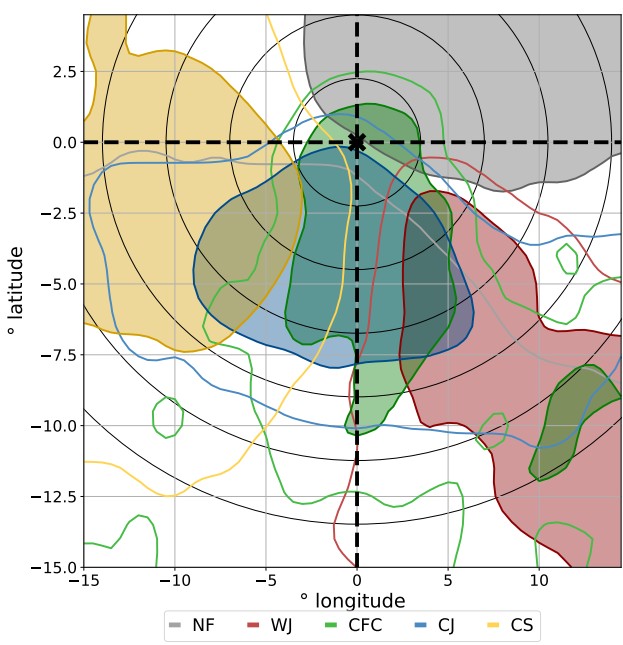

**Figure B3.** As Fig. 7 but divided by the number of windy conditions.

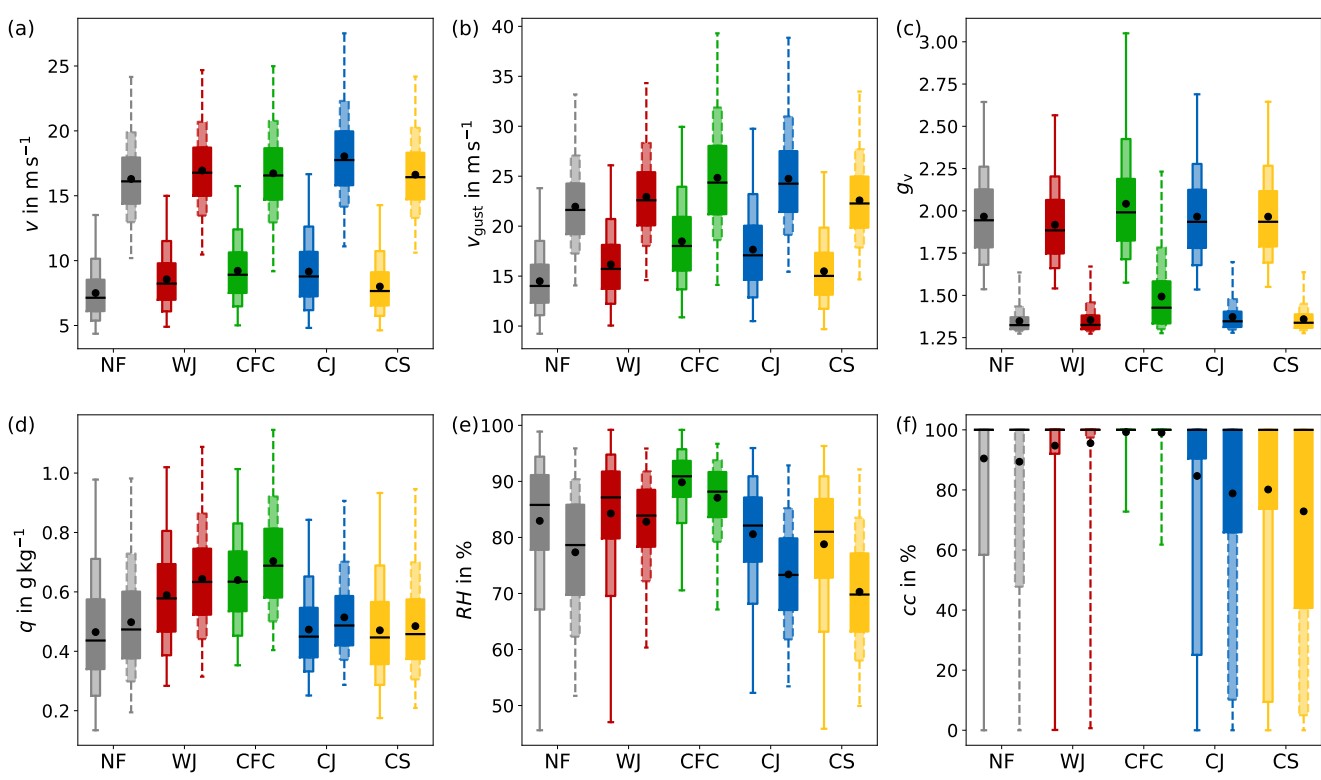

**Figure B4.** As Fig. 9 but with land (solid boxplots) and sea (dashed boxplots) grid points separated.

*Author contributions.* LE compiled and evaluated the climatology and wrote the original manuscript. BS computed the feature probabilities and gave advise on the statistical analysis. PK designed the overall project, acquired the funding and coordinated the scientific work. PK and JGP jointly supervise the PhD of LE. All authors contributed to discussions and text revisions.

*Competing interests.* At least one of the (co-)authors is a member of the editorial board of *Weather and Climate Dynamics*. The peer-review process was guided by an independent editor, and the authors also have no other competing interests to declare.

*Acknowledgements.* The research leading to the results in this paper has been accomplished within the "Dynamical feature-based ensemble postprocessing of wind gusts within European winter storms" C5 project of the "Waves to Weather" Transregional Collaborative Research Center (grant no. SFB/TRR 165) funded by the German Science Foundation (DFG). JGP thanks the AXA Research Fund for support. The authors are grateful to Sebastian Trepte (DWD) and Olivier Mestre (Météo France) for providing the surface observations data set, to Robert Redl (LMU) for preprocessing the data, and to Ting-Chen Chen (KIT) for the computation of the cyclone tracks and helpful scripts. Further
thanks go to Sebastian Lerch (KIT) for many useful discussions and helpful comments in the framework of C5. The authors also acknowledge Suzanne L. Gray and an anonymous reviewer for their valuable comments, which helped to improve this article.

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
