# Peer review of "Identification of high-wind features within extratropical cyclones using a probabilistic random forest - Part 2: Climatology over Europe"

_Weather and Climate Dynamics, 2023_

## Referee Comment (RC1)

**Review of 'Identification of high-wind features within extratropical cyclones using a probabilistic random forest - Part 2: Climatology' by Eisenstein et al. submitted to Weather and Climate Dynamics**

**General comments:**

This is an interesting study describing the results of applying a methodology previously published by the authors to develop a 19-extended-winters climatology of mesoscale wind features in cyclones over Western and Central Europe. The topic is very suitable for this journal. The research is novel in the generation of this type of climatology over this region, and in the method used to obtain the climatology. I have a few questions about the details of the methods (some of the description is confusing, at least to me) and about the robustness of the results. The work would also benefit from a more detailed comparison with previous literature to draw out the novelties of the findings. The paper is well presented apart from some small language glitches (not all of which I've likely commented on in my review).

**Major specific comments:**

**L105** Here it is stated that the focus of this study is on cyclones occurring in Western and Central Europe. This geographical focus of the study should be included in the paper title.

**Introduction** Given the focus of this paper on the different sources of strong low-level winds, it might be helpful to include a diagram showing where these are located within a classic cyclone as a first figure. The different features are explained in the text but it may be difficult for less knowledgeable readers to visualise these. Related to this comment, some more detail on how the features are defined in probabilistic characterisation would be helpful. I realise that these details are described in the part 1 of this paper but it should be possible for readers to follow part 2 without having first read part 1.

**L73** Here you refer to the Earl et al. study. One interesting result from that study is the different of the prevelence of the features changes depending on whether those generating the top 1% or 0.1% of daily maximum wind gusts are considered. Did you consider changing your windspeed threshold to look at how the relative importance of the features changes?

**L203** "the proportion of NF is considerably lower in COSMO-REA6 data" – this statement is true but "no feature" strong winds still account for about 30% of the strong winds. Please can you say more about what these strong winds are likely to be caused by and whether they are likely to be linked to cyclones? How often are these NF strong wind regions close to the tracked cyclones (all strong wind regions are within 15 degrees of a cyclone (L167), but are the NF regions relatively more likely to be further away (e.g. beyond 7.5 degrees)?

**Section 3.1** Here the pie charts showing the relative frequency of the different strong wind features are discussed with the pie charts showing percentage occurrence values to 1 decimal place. It would be useful to have some sense of how robust these relative percentages are to small changes in the datasets used and hence how meaningful the interpretations are. For example, what happens if some of the years are omitted from the datasets used, the definition of strong winds is changed slightly, or the domain of interest (the blue box in Fig. 2) is shifted slightly?

**L228** For calculation of the seasonal and inter-annual evolution of the wind features, the strong winds need to be present at at least 100 grid points over the domain. The model grid is a latitude-longitude grid with a spacing of 0.0625 degrees. Hence, the size of the grid boxes (in km$^2$) changes with latitude. Is this change taken into account?

**Existing literature** While I don't know of other papers that have produced long term climatologies of mesoscale wind features, there are some other papers that the authors should consider referencing. Rivière et al. (2014, https://rmets.onlinelibrary.wiley.com/doi/10.1002/qj.2412) shows the evolution from a dominant warm to dominant cold conveyor belt jet (see fig. 10 particularly) and Dacre et al.

(2012, https://journals.ametsoc.org/view/journals/bams/93/10/bams-d-11-00164.1.xml) describes the generation of a cyclone "atlas" which includes analysis of the warm and cold conveyor belts. I also suggest attempting a more quantitative comparison with the existing literature that is cited. The term "mostly consistent" is used in the conclusions – can you provide a deeper comparison? How do the cyclones over the Western and Central Europe region analysed here compare with those over the North Atlantic or UK? It would also be helpful to contrast the descriptions of the cold and warm jets here with some much older literature. For example, for me the classic paper on cyclone structure is Browning and Roberts (1994, https://rmets.onlinelibrary.wiley.com/doi/epdf/10.1002/qj.49712052006). See also Browning (2005, https://rmets.onlinelibrary.wiley.com/doi/10.1256/qj.03.201). This deeper analysis would help the reader understand what new knowledge is generated by this research. Currently this seems rather modest as it is detailed in just two sentences in the conclusions as follows: "The large number of storms investigated helped revealing the large variability in the location of CFC in a system-relative framework. Other differences to the literature include the time of occurrence of the CJ already several hours before the time of maximum depth in contrast to Hewson and Neu (2015)."

**Minor specific comments:**

**L2** I suggest replacing "high winds" with "strong winds" as used on the previous line as "high" could refer to the altitude of the winds rather than their strength. Please also consider other places where "high" or "highest" is used.

**L3** "strong cold-sector winds" are not a "dynamical feature" (unlike the other mesoscale features considered).

**L35** Here it is stated that cold frontal convection is more common in cyclones following the Norwegian cyclone model. Can you point to evidence supporting this statement? Although cyclones following the Shapiro-Keyser conceptual model undergo frontal fracture, the cold front can be stronger than the warm front in idealised lifecycle simulations of these cyclones with the converse true for Norwegian-type cyclones (e.g., see Fig. 3 of Shapiro et al. 1999: https://link.springer.com/chapter/10.1007/978-1-935704-09-6_14).

**L55** Regarding tracking cyclones over the Mediterranean, you might be interested in this recent paper: https://wcd.copernicus.org/preprints/wcd-2022-63/

**L59** Probably worth pointing out that the calculation of Laurila et al. considers the wind extremes using monthly values (i.e., the extreme wind factor is the monthly 98th percentile/monthly mean)

**L66** In this paragraph you should point out that the studies referred to consider different regions e.g., the Parton et al. and Earl et al. studies only consider the UK and so may not be representative of cyclones in general.

**L107** Here it states that "January 2001 to December 2019" data is used whereas at the start of this section the observational dataset is described as extending to mid-2020.

**L114** "we concentrate here on October 2000 to March 2019, i.e., a minor shift of three months". The model data starts 3 months earlier than the observational data but ends 9 months earlier. The reason this doesn't make any difference is because only extended winter data is considered - this could be made clearer.

**L114** Can you please state the number of data points used to calculate the 98th percentile at each location and date. I think it's 210 (21 days of data for each of 10 years) but it would be good to know if I'm correct. The 98th percentile winds at any grid point will be strongly dependent on whether a localised strong wind region happens to cross that grid point on one of these 210 days. How does the value of this 98th percentile (or the lower "windy" threshold) vary spatially for a sample time snapshot? Also, is it correct that the wind speed threshold at a given time and place will be different in absolute terms for the model and observational data because the normalisation is calculated separately for the model and observational data?

**L129** Here it states "Here, we apply RAMEFI to station observations and COSMO-REA6 data under windy conditions during the extended winter months, regardless of whether a storm occurred or not. However, we later filter the output for cyclone occurrence as discussed in Sect. 2.4." Please can you clarify if any of the results shown are not filtered for cyclone occurrence? I wondered if the filtering for cyclones might only be applied in Sections 3.2 and 3.3 or even just in Section 3.3.

**Fig. 2** Can you provide more details on how the track density is calculated? This field is very smooth in the plot so I suspect that there is some averaging occurring (typically track densities are plotted as the number of tracks within a certain area, such as a 5 degree spherical cap, of each grid point). I don't think it's possible that there are up to about 3 storms with their MSLP centre located at a single grid point each winter. Also, it would be more helpful to readers for the track density per year to be plotted rather than the number over the 19 years.

**Fig. 3** The domain used for analysis of the observational data is, I think, smaller than that used for the model data (see L105 and L118) with the latter extending further east and north. However, a further domain is specified in L169. If these domains are different then how does this difference affect the comparison between the results from the observations and model output shown in this figure? What would the figure look like if the same domains were used?

**Fig. 4** Does this plot consider a feature as occurring if it has at least 100 grid points with any percentage of likelihood at each grid point or only if it is the most probable feature at a given grid point? What is the definition of a stormy time step (is it that any feature is present at at least 100 grid points?).

**L242** "However, the recent two decades w[h]ere characterised by an unusual number of storms in October," - please provide evidence for this statement.

**L243** I'm confused by the statement "given the lower threshold of $\tilde{v}$ - leads to a larger number of stormy time steps in the 19 years". $\tilde{v}$ is the wind speed normalised by it's 98th percentile value on the day of the year (and time of day and location) over the period 2005-2015 and stormy timesteps are those in which the windspeed exceeds 80% of $\tilde{v}$. Given this normalisation I'd expect the number of stormy days not to vary too much between months (as that is what the normalisation is trying to achieve). I suspect I've misunderstood something here - is it that the location specific nature of the windspeed threshold simply means that it picks out all stormy events? Please can you explain?

**L247** Here the frequency of the cold jet is linked to that of cold air outbreaks. Please can you explain why these two phenomena could be expected to be linked?

**L271** The definition of "windy" is different here to that in L167. L167 defines it as strong winds "in the vicinity of 15 degrees of the cyclone centre," which implies a circular region around the cyclone whereas here a box with ±15 degrees in the zonal direction and -15 to 5 degrees in meridional direction is stated. Which is correct? Also, how is the reducing zonal extent (in km) with latitude accounted for (e.g., 30 degrees in the zonal direction varies from 2330 to 1390 km between 45 and 65 degrees North)?

**299** Here it is stated "As orography can induce convection, CFC might be detected without the occurrence of a cold front or even cyclone". I thought only windy features were identified close to a cyclone were considered though (see above comment and also the comment relating to line 129).

**Fig. 6** I think a constant latitude has been assumed here (of 50 degrees) in the calculation of the distance circles as they appear circular. If the variation of zonal distance with latitude had been considered the distance circles would not be circular. This assumption should be stated in the caption. Please also add to the caption that the grey shading in (a) is for the NF winds.

**Fig. 6a** How is the location of the NF winds close to the cyclone centre consistent with Fig. 5b which shows NF winds are typically found to the south and north of the main storm track?

**L373** "RR values over $1\,\mathrm{m\,s^{-1}}$". "RR" isn't defined in this paper but based on the earlier paper by the authors I guessed that it stood for precipitation amount. However, the units are then incorrect. Please define RR.

**Fig. 8** What heights are all the fields considered at?

**L383** Here it is stated that it is not surprising that the highest gust ratio is found for the CFC features because convection is associated with high instability and turbulence. The grid spacing of the model is about 6 km. Does it use a convection scheme or is convection represented explicitly? How are the gusts diagnosed in the model? Related to this point, it would be interesting to know how the climatology of model gusts shown in Section 4 compares to a similar climatology generated from the observations given the likely parametrization dependence of the model-derived gusts. Have you considered this?

**L425** What is the evidence from your analysis that the CFC feature requires a convective trigger? The term "trigger" is also vague: do you include both forced ascent and ascent due to the release of convective instability?

**Technical errors:**

**L34** "causing" $\rightarrow$ "it causes".

**L44** "As CJ" $\rightarrow$ "As the CJ"

**L51** "perception on" $\rightarrow$ "perception of"

**L74** "convection-induces" $\rightarrow$ "convection-induced"

**L85** "northeastern" $\rightarrow$ "northeast"

**L106, L170, L190** "less than" $\rightarrow$ "fewer than" (n.b. you should use "fewer than" if you can count the objects e.g., fewer bottles of lemonade but less lemonade). Please also check elsewhere in the paper.

**L110** "Ger- man" $\rightarrow$ "German"

**L156** Remove full stop before start of 1st sentence.

**L168** "in altitudes above 800m consistent to Part 1" $\rightarrow$ "at altitudes above 800m consistent with Part 1"

**L197** "perscpective" - spelling.

**L232** "any" $\rightarrow$ "every"

**L239** "amount" $\rightarrow$ "number" (because the number of stormy timesteps can be counted).

**L241** "consistent to Feser et al." → "consistent with Feser et al."

**L242** "where" → "were"

**L244** "corresponds" → "correspond"

**L301** "of CJ" → "of the"

**L306** add "this" before "suggests"

**L326** "(Gentile and Gray, 2023)" → "Gentile and Gray (2023)".

**L330** "particular" → "particularly"

**L446** "helped revealing" → "helped to reveal"

**L450** "larger" → "longer"

---

## Author Comment (AC3)

**Review of 'Identification of high-wind features within extratropical cyclones using a probabilistic random forest - Part 2: Climatology' by Eisenstein et al. submitted to Weather and Climate Dynamics**
**10.5194/wcd-2023-10**

**General comments:**

This is an interesting study describing the results of applying a methodology previously published by the authors to develop a 19-extended-winters climatology of mesoscale wind features in cyclones over Western and Central Europe. The topic is very suitable for this journal. The research is novel in the generation of this type of climatology over this region, and in the method used to obtain the climatology. I have a few questions about the details of the methods (some of the description is confusing, at least to me) and about the robustness of the results. The work would also benefit from a more detailed comparison with previous literature to draw out the novelties of the findings.

The paper is well presented apart from some small language glitches (not all of which I've likely commented on in my review).

Thank you for the thorough review of our manuscript and many valuable comments that help to improve our paper. We have been carefully considering each comment, which are addressed in blue below. Text changes and more details will be provided in the Author's response.

**Major specific comments:**

**L105** Here it is stated that the focus of this study is on cyclones occurring in Western and Central Europe. This geographical focus of the study should be included in the paper title.
Thank you for raising this. We add "over Europe" at the end of the title.

**Introduction** Given the focus of this paper on the different sources of strong low-level winds, it might be helpful to include a diagram showing where these are located within a classic cyclone as a first figure. The different features are explained in the text but it may be difficult for less knowledgeable readers to visualise these. Related to this comment, some more detail on how the features are defined in probabilistic characterisation would be helpful. I realise that these details are described in the part 1 of this paper but it should be possible for readers to follow part 2 without having first read part 1.
We understand the concern but given the easy and open access of Part 1 we do not think it is necessary to reuse its Figure 1 in Part 2. However, we specifically refer to Figure 1 of Part 1 in the Introduction now and add a few sentences about the distinction of the features in Section 2.3

**L73** Here you refer to the Earl et al. study. One interesting result from that study is the different of the prevelence of the features changes depending on whether those generating the top 1% or 0.1% of daily maximum wind gusts are considered. Did you consider changing your windspeed threshold to look at how the relative importance of the features changes?
This is indeed an interesting result to add. Furthermore, we add pie diagrams with higher wind thresholds and a corresponding discussion to Section 3.1 showing an increase of CJ and CFC occurrence with higher thresholds (see response to Section 3.1 below).

**L203** "the proportion of NF is considerably lower in COSMO-REA6 data" – this statement is true but "no feature" strong winds still account for about 30% of the strong winds. Please can you say more about what these strong winds are likely to be caused by and whether they are likely to be linked to cyclones? How often are these NF strong wind regions close to the tracked cyclones (all strong wind regions are within 15 degrees of a cyclone (L167), but are the NF regions relatively more likely to be further away (e.g. beyond 7.5 degrees)?
NF winds are commonly associated with weaker winds (see our response to your next comment and the added Fig. FR2) and, as already mentioned in the preprint, NF detects parts of the CCBa. Looking at a relative version of the system-relative composite (number of occurrences divided by the number of windy conditions → Figure FR1), the contours of NF shift to the north-eastern quadrant,

outside the area of the other features, which show less change. We include this figure in the Appendix.

Furthermore, Part 1 showed that for some cases the spatial independence leads to uncertainty in the area where the introduced features occur, resulting in NF being the most probable features (see Section 7.4 of Part 1 for more details) or the area between double fronts shows uncertainty (see Section 7.1). This is not commonly the case; however, it might explain the overlap in the southern quadrants to some extent.

[Figure]

*Figure FR1: As Fig. 6 but divided by the number of windy conditions.*

**Section 3.1** Here the pie charts showing the relative frequency of the different strong wind features are discussed with the pie charts showing percentage occurrence values to 1 decimal place. It would be useful to have some sense of how robust these relative percentages are to small changes in the datasets used and hence how meaningful the interpretations are. For example, what happens if some of the years are omitted from the datasets used, the definition of strong winds is changed slightly, or the domain of interest (the blue box in Fig. 2) is shifted slightly?

This is a valid point, and we had a further look into this. First, we add pie diagrams for higher wind thresholds to the main text (Figure FR2) showing that the proportion of NF decreases significantly, while especially the proportions of CJ and CFC increase consistently to Earl et al. (2017). Furthermore, three sets of ten randomly chosen winter seasons show only slight changes in proportions (Figure FR3), which is expected given the relatively small fluctuations shown in Fig. 4. A shift in the spatial domain would yield different proportions as can be seen in the Earth-relative plot in the preprint (Fig. 5).

[Figure]

*Figure FR2: As Fig. 2 m,o (COSMO-REA6 all; MAXP) but for grid points with (a),(b) $\tilde{v} \geq 1.2$ and (c),(d) $\tilde{v} \geq 1.4$.*

[Figure]

*Figure FR3: As Fig. 2 m--o (COSMO-REA6 all) but for three different sets of 10 randomly chosen winter seasons.*

**L228** For calculation of the seasonal and inter-annual evolution of the wind features, the strong winds need to be present at at least 100 grid points over the domain. The model grid is a latitude-longitude grid with a spacing of 0.0625 degrees. Hence, the size of the grid boxes (in km2) changes with latitude. Is this change taken into account?

The choice of 100 grid points is subjective and was chosen to remove noise while keeping the rather small-scale CFC feature. This could have been based on an area threshold, too, but given that our data are on a lat-lon grid, we decided to use that. See further discussion below (comment to L271).

**Existing literature** While I don't know of other papers that have produced long term climatologies of mesoscale wind features, there are some other papers that the authors should consider referencing. Rivière et al. (2014, https://rmets.onlinelibrary.wiley.com/doi/10.1002/qj.2412) shows the evolution from a dominant warm to dominant cold conveyor belt jet (see fig. 10 particularly) and Dacre et al. (2012, https://journals.ametsoc.org/view/journals/bams/93/10/bams-d-11-00164.1.xml) describes the generation of a cyclone "atlas" which includes analysis of the warm and cold conveyor belts. I also suggest attempting a more quantitative comparison with the existing literature that is cited. The term "mostly consistent" is used in the conclusions – can you provide a deeper comparison? How do the cyclones over the Western and Central Europe region analysed here compare with those over the North Atlantic or UK? It would also be helpful to contrast the descriptions of the cold and warm jets here with some much older literature. For example, for me the classic paper on cyclone structure is Browning and Roberts (1994, https://rmets.onlinelibrary.wiley.com/doi/epdf/10.1002/qj.49712052006). See also Browning (2005, https://rmets.onlinelibrary.wiley.com/doi/10.1256/qj.03.201). This deeper analysis would help the reader understand what new knowledge is generated by this research. Currently this seems rather modest as it is detailed in just two sentences in the conclusions as follows: "The large number of storms investigated helped revealing the large variability in the location of CFC in a system-relative framework. Other differences to the literature include the time of occurrence of the CJ already several hours before the time of maximum depth in contrast to Hewson and Neu (2015)."

Thank you for the suggested references. A more thorough comparison with literature will be implemented in the introduction and conclusions but also throughout the paper when relevant.

**Minor specific comments**:

**L2** I suggest replacing "high winds" with "strong winds" as used on the previous line as "high" could refer to the altitude of the winds rather than their strength. Please also consider other places where "high" or "highest" is used.

While we understand this concern, we think the usage of "high" synonymously to "strong" is clear in the paper. We change this in some places but decided to not change this throughout the paper to be consistent to Part 1 (and the title).

**L3** "strong cold-sector winds" are not a "dynamical feature" (unlike the other mesoscale features considered).

Thank you for drawing attention to this. We rephrase this sentence.

**L35** Here it is stated that cold frontal convection is more common in cyclones following the Norwegian cyclone model. Can you point to evidence supporting this statement? Although cyclones following the Shapiro-Keyser conceptual model undergo frontal fracture, the cold front can be stronger than the warm front in idealised lifecycle simulations of these cyclones with the converse true for Norwegian-type cyclones (e.g., see Fig. 3 of Shapiro et al. 1999: https://link.springer.com/chapter/10.1007/978-1-935704-09-6 14).

Several Shapiro-Keyser cyclones in the recent years showed a less active cold front (e.g., Egon 2017, Xavier 2017, Friederike 2018). This is, however, not directly linked to a "weaker" cold front. We realise that we cannot provide statistical evidence such that we rephrase this statement to clarify this.

**L55** Regarding tracking cyclones over the Mediterranean, you might be interested in this recent paper: https://wcd.copernicus.org/preprints/wcd-2022-63/

Thank you for the recommendation.

**L59** Probably worth pointing out that the calculation of Laurila et al. considers the wind extremes using monthly values (i.e., the extreme wind factor is the monthly 98th percentile/monthly mean)

Thank you for pointing this out. We clarify this in the text.

**L66** In this paragraph you should point out that the studies referred to consider different regions e.g., the Parton et al. and Earl et al. studies only consider the UK and so may not be representative of cyclones in general.

This is done accordingly.

**L107** Here it states that "January 2001 to December 2019" data is used whereas at the start of this section the observational dataset is described as extending to mid-2020.

Although the data set is available until mid-2020, we do not include 2020 to get the same number (19) for each month and being as close to the COSMO-REA6 data set as possible. We clarify this in the text. See also next comment.

**L114** "we concentrate here on October 2000 to March 2019, i.e., a minor shift of three months". The model data starts 3 months earlier than the observational data but ends 9 months earlier. The reason this doesn't make any difference is because only extended winter data is considered - this could be made clearer.

We apologise for the confusion. The COSMO-REA6 data set is actually available from 1995 to mid-2019, while the observations are available from 2001 to mid-2020. To make the data sets as comparable as possible, we include 19 extended winter seasons with the same number of months, however, a slight shift of three months. We clarify this in the text.

**L114** Can you please state the number of data points used to calculate the 98th percentile at each location and date. I think it's 210 (21 days of data for each of 10 years) but it would be good to know if I'm correct. The 98th percentile winds at any grid point will be strongly dependent on whether a

localised strong wind region happens to cross that grid point on one of these 210 days. How does the value of this 98th percentile (or the lower "windy" threshold) vary spatially for a sample time snapshot? Also, is it correct that the wind speed threshold at a given time and place will be different in absolute terms for the model and observational data because the normalisation is calculated separately for the model and observational data?

210 data points is correct for COSMO-REA6. For Part 1 we tested the computation with +-30 days for an exemplary month but could not see substantial differences. To save computational costs we decided to keep +-10 days instead.

Figure FR4 shows a snapshot for COSMO-REA6 (left) and overlayed with observations (right). While only including 10 years – instead of 20 as done for the observational data set – might give exceptionally strong events more weight, the comparison of COSMO-REA6 to observations shows only slight differences, which are probably due to the different data sets instead of single events. The biggest differences occur at mountain peaks as to be expected.

[Figure]

*Figure FR4: 98th percentile of wind speed in m/s for 01 January, 0 UTC. COSMO-REA6 (left) and COSMO-REA6 overlayed with observations (right).*

**L129** Here it states "Here, we apply RAMEFI to station observations and COSMO-REA6 data under windy conditions during the extended winter months, regardless of whether a storm occurred or not. However, we later filter the output for cyclone occurrence as discussed in Sect. 2.4." Please can you clarify if any of the results shown are not filtered for cyclone occurrence? I wondered if the filtering for cyclones might only be applied in Sections 3.2 and 3.3 or even just in Section 3.3.

We apologise for the confusion. All results are filtered by cyclone and area to be consistent throughout the paper.

**Fig. 2** Can you provide more details on how the track density is calculated? This field is very smooth in the plot so I suspect that there is some averaging occurring (typically track densities are plotted as the number of tracks within a certain area, such as a 5 degree spherical cap, of each grid point). I don't think it's possible that there are up to about 3 storms with their MSLP centre located at a single grid point each winter. Also, it would be more helpful to readers for the track density per year to be plotted rather than the number over the 19 years.

The track density calculation considers a radius of 750km. We calculate the sum of previously identified cyclone tracks in any grid points that are at least once within the domain. The track density is then the total accumulated number of tracks divided by the given time period to get the track density per year. This will be clarified in the text and caption accordingly.

**Fig. 3** The domain used for analysis of the observational data is, I think, smaller than that used for the model data (see L105 and L118) with the latter extending further east and north. However, a further domain is specified in L169. If these domains are different then how does this difference affect the comparison between the results from the observations and model output shown in this figure? What would the figure look like if the same domains were used?

First of all, we found a mistake in the manuscript. The analysed zonal extent for COSMO-REA6 is 10°W to 25°E (not 30°E). Figure FR5 shows pie diagrams for observations (top row) and COSMO-

REA6 land grid points (lower row) for the same domain (10°W to 20°E, 45°N to 60°N) showing just slight differences to the original pie diagrams in Fig. 3.

[Figure]

*Figure FR5: As Fig 2a—h for observations (top row) and COSMO-REA6 grid points over land (bottom row) for a consistent area of 10°W to 20°E, 45°N to 60°N. The first and third column show MAXP, while the second and fourth column show ACCP.*

**Fig. 4** Does this plot consider a feature as occurring if it has at least 100 grid points with any percentage of likelihood at each grid point or only if it is the most probable feature at a given grid point? What is the definition of a stormy time step (is it that any feature is present at at least 100 grid points?).

It is considered when it is the most probable feature. We added this to the text.
The definition of stormy time steps is written in l. 171f but will be clarified.

**L242** "However, the recent two decades w[h]ere characterised by an unusual number of storms in October," - please provide evidence for this statement.

This statement arised from a submitted paper (Mömken et al. 2023). Looking at insurance data and SSI (at least over Germany), we found almost no noteworthy winter storms in November in our time period while we found several in October (e.g., Christian/St. Jude's Day Storm 2013, Xavier 2017, Herwart 2017). We rephrased this statement clarifying this.

**L243** I'm confused by the statement "given the lower threshold of $\tilde{v}$ - leads to a larger number of stormy time steps in the 19 years". $\tilde{v}$ is the wind speed normalised by it's 98th percentile value on the day of the year (and time of day and location) over the period 2005-2015 and stormy timesteps are those in which the windspeed exceeds 80% of $\tilde{v}$. Given this normalisation I'd expect the number of stormy days not to vary too much between months (as that is what the normalisation is trying to achieve). I suspect I've misunderstood something here - is it that the location specific nature of the windspeed threshold simply means that it picks out all stormy events? Please can you explain?

If the threshold was the 80th percentile of v, the number of exceedances would be the same over the ocean and over land. However, we consider 80% of the 98th percentile. As seen in Figure A1, $v_{98}$ is about twice as high for ocean grid points and the area between 80% of $v_{98}$ is larger for ocean grid points compared to land. As written in Appendix A, ocean grid points experience windy conditions around 45% more often than land grid points. Therefore, a cyclone might be considered a "storm" over the ocean, but not once it reaches land.
We realise that "lower" is misleading here and we will rephrase this sentence.

**L247** Here the frequency of the cold jet is linked to that of cold air outbreaks. Please can you explain why these two phenomena could be expected to be linked?

The two phenomena are not directly linked and our statement is indeed misleading. We will rephrase and clarify this.

**L271** The definition of "windy" is different here to that in L167. L167 defines it as strong winds "in the vicinity of 15 degrees of the cyclone centre," which implies a circular region around the cyclone whereas here a box with ±15 degrees in the zonal direction and -15 to 5 degrees in meridional direction is stated. Which is correct? Also, how is the reducing zonal extent (in km) with latitude accounted for (e.g., 30 degrees in the zonal direction varies from 2330 to 1390 km between 45 and 65 degrees North)?

We apologise for the confusion. For the whole climatology, we considered ±15° in the zonal direction and -15 to 5° in meridional direction as commented above.

As mentioned above, the changing distance in km for different latitudes was not considered, as our data is on a lat-lon grid. This is similar to other papers using cyclone-relative analysis (e.g., Sinclair et al., 2020 (https://wcd.copernicus.org/articles/1/1/2020/), Priestley et al., 2022 (https://wcd.copernicus.org/articles/3/337/2022/)). We do not anticipate that a change to area-thresholds would change the overall statements in our discussion. It would be desirable to test this effect, but the computational costs to do this for the whole climatology are quite high.

**299** Here it is stated "As orography can induce convection, CFC might be detected without the occurrence of a cold front or even cyclone". I thought only windy features were identified close to a cyclone were considered though (see above comment and also the comment relating to line 129).

This is indeed misleading. What we wanted to say is that wind speeds not associated with one of the features but for example associated with a strong pressure gradient hitting orography can induce convection. For example, this commonly happens over the Norwegian mountains.

**Fig. 6** I think a constant latitude has been assumed here (of 50 degrees) in the calculation of the distance circles as they appear circular. If the variation of zonal distance with latitude had been considered the distance circles would not be circular. This assumption should be stated in the caption. Please also add to the caption that the grey shading in (a) is for the NF winds.

Both will be clarified in the caption.

**Fig. 6a** How is the location of the NF winds close to the cyclone centre consistent with Fig. 5b which shows NF winds are typically found to the south and north of the main storm track?

The histograms in Fig. 5 are divided by the number of windy conditions. This is not done in Fig. 6a. Doing so leads to NF mostly occurring in the north-eastern quadrant as discussed above (see Fig. FR1).

**L373** "RR values over 1m s−1". "RR" isn't defined in this paper but based on the earlier paper by the authors I guessed that it stood for precipitation amount. However, the units are then incorrect. Please define RR.

Precipitation amount $RR$ has been introduced in Section 2.1 (l99). However, we apologise for the confusion regarding the units. These are of course mm h$^{-1}$.

**Fig. 8** What heights are all the fields considered at?

We added the height in Section 2 (Data and methods): mean sea level pressure ($p$), 2m air temperature ($T$), wind speed at 10m ($v$), wind direction at 10m ($d$) and precipitation amount ($RR$).

**L383** Here it is stated that it is not surprising that the highest gust ratio is found for the CFC features because convection is associated with high instability and turbulence. The grid spacing of the model is about 6 km. Does it use a convection scheme or is convection represented explicitly? How are the gusts diagnosed in the model? Related to this point, it would be interesting to know how the climatology of model gusts shown in Section 4 compares to a similar climatology generated from the observations given the likely parametrization dependence of the model-derived gusts. Have you considered this?

The model uses the mass flux scheme from Tiedtke (1989) for subgrid-scale convection. A note on this will be added in Section 2.2.

Unfortunately, the observational data set we use does not include wind gusts, such that a comparison is not possible.

**L425** What is the evidence from your analysis that the CFC feature requires a convective trigger? The term "trigger" is also vague: do you include both forced ascent and ascent due to the release of convective instability?

This statement indeed includes poor choices of words. It will be rephrased and clarified.

**Technical errors:**

Thank you for making us aware of these errors. All will be changed accordingly.

L34 "causing" → "it causes".

L44 "As CJ" → "As the CJ"

L51 "perception on" → "perception of"

L74 "convection-induces" → "convection-induced"

L85 "northeastern" → "northeast"

L106, L170, L190 "less than" → "fewer than" (n.b. you should use "fewer than" if you can count the objects e.g., fewer bottles of lemonade but less lemonade). Please also check elsewhere in the paper.

L110 "Ger- man" → "German"

L156 Remove full stop before start of 1st sentence.

L168 "in altitudes above 800m consistent to Part 1" → "at altitudes above 800m consistent with Part 1"

L197 "perscpective" - spelling.

L232 "any" → "every"

L239 "amount" → "number" (because the number of stormy timesteps can be counted).

L241 "consistent to Feser et al." → "consistent with Feser et al."

L242 "where" → "were"

L244 "corresponds" → "correspond"

L301 "of CJ" → "of the"

L306 add "this" before "suggests"

L326 "(Gentile and Gray, 2023)" → "Gentile and Gray (2023)".

L330 "particular" → "particularly"

L446 "helped revealing" → "helped to reveal"

L450 "larger" → "longer"

---

## Author Comment (AC4)

'Review of "Identification of high-wind features within extratropical cyclones using a probabilistic random forest – Part 2: Climatology" by Lea Eisenstein et al.'
10.5194/wcd-2023-10

This paper provides a valuable 19-year, extended-winter climatology of strong winds related to cyclones in northwestern Europe. The novel aspect is the use of the RF-based approach recently published by the authors to label the near-surface winds according to distinct mesoscale features that produce the strong winds: the warm jet, cold jet, cold-front convection and cold-sector winds, along with a 'no feature' category. This distinction indeed shows different Earth-relative and cyclone-relative characteristics, and it opens the door to understanding the predictability of cyclone-induced winds which would likely depend on the feature.
The scope fits WCD well, the text is very well written and the figures compelling. I enjoyed reading the manuscript and I overall support its progress towards publication. There are, however, some aspects needing attention and clarification before its acceptance.

Thank you for the thorough review of our manuscript and many valuable comments that help to improve our paper. We have been carefully considering each comment, which are addressed in blue below. Text changes and more details will be provided in the Author's response.

**Major comments**
1.  Some key aspects related to the RAMEFI method need to be enhanced in the manuscript, so that Part 2 can stand alone: please explain how can station data from only 12 cyclones construct a sufficient training dataset for its applicability to (i) all cyclones in the climatology, to (ii) other, gridded data, and to (iii) location-specific features, such as land/sea differences or orographic effects.
    We will expand the description of RAMEFI regarding the following aspects:
    (i) While 12 case studies seem to be quite few, they represent a healthy diversity of cyclone developments and features including very intense and more moderate cyclones with differing storm tracks. Overall, almost 280 time steps leading to almost 80,000 data points are included in training the RF. The promising evaluation using a cross-validation approach in Part 1 suggests a reliable detection of the features in long-term data for most cyclones
    (ii) The statistical evaluation of the application on COSMO-REA6 data in Part 1 demonstrates that RAMEFI generates reliable identification of the features for gridded data despite being trained solely on surface observations.
    (iii) While the normalisation of $v$ and $\theta$ brings some independence of location-specific effects and we remove grid points/stations over 800m, some areas are still affected by nearby orography (e.g., Norway and the Balkans) and results should be treated with some caution here. As written in the manuscript, we did not statistically evaluate RAMEFI over the ocean and smaller differences are to be expected, but looking at several case studies did not show any obvious issues.

2.  It is not entirely clear how the distinction is made among the WJ/CJ/CS/CFC during the initial subjective labelling in the vicinity of the front. Since the front is a narrow and slanted surface, the location of the front line changes with height, and accordingly the attribution of the windy grid points on both its sides. It is unclear how the station-based identification of convection relates to the commonly-used front detection by the 850-hPa temperature/wind/humidity gradients and why these features are mostly absent from oceanic locations. In this context I find the comment in lines 299-300 somewhat disturbing. How should one distinguish between winds caused by frontal convection from any other isolated convective downdrafts? It would be beneficial to clarify these points, so that the high uncertainty of the CFC winds can be reduced or better understood.
    Please note that we do not identify the front itself, but the convection induced in association with a front. While the most important predictor is $RR$, further predictors ensure the location in the frontal region, such that post-CFC is identified as CS rather than CFC. These include for example $\Delta p$ and $\theta$. However, in orographic regions such as the Norwegian mountains, CFC can be falsely detected.

3. The description of the application of the RAMEFI method to both station and model data should be enhanced. It was not clear to me from the text whether the method is carried out independently for each data set, and therefore providing independent probabilities for each, and why is it designed so? Therefore I wasn't sure if the differences among the datasets seen in Fig. 3 stem from the fact that there are stations only over land, and they are not evenly distributed, or whether there are additional biases from the application of the method to each dataset separately?

RAMEFI makes decision based on only the single station or grid point at a certain time. Hence, it is not only applied to the data sets individually, but the output is also not dependent on neighbouring stations or grid points.

Small biases are to be expected as the evaluation of COSMO-REA6 is slightly worse than for the observation data set, it was trained on. This is the reason why we compare the two data sets in this section. More substantial differences are rather due to data point distribution and overall differences in the development of a cyclone in the data sets.

4. The fact that NF is not randomly distributed in Figs. 6 and 7 suggests that some wind is currently associated to NF even though it should be coherently linked to the cyclone and is therefore mislabeled. Please quantify the sensitivity of the results to the labelling procedure, and estimate how much of the NF signal in Figs. 3 or 5 is found in the vicinity of cyclones.

Fig. 6 and 7 show the absolute occurrence, i.e., independently of the other features or number of windy data points. The south-western area of a cyclone is characterised by higher wind speeds, while the north-eastern quadrant shows windy conditions less often. The relative occurrence shows the NF feature mostly in the north-eastern region as shown in Fig. FR1, which will be added to the Appendix.

This climatology showed us that for example the CCBa, i.e., the CJ before it wraps around the cyclone centre, is more common than we expected when developing the method as a distinct peak of NF occurrence is visible there. Although, the CCBa could be considered as an additional feature in future work, the focus here is on WJ, CJ, CS and CFC, which are the most common causes of high-winds, and their detection is not hindered by CCBa being included in NF.

The absolute occurrence of NF in the south-western quadrant is mostly caused by uncertainty, e.g., due to double fronts or untypical structures (see Sections 7.1 and 7.3 in Part 1).

[Figure]

*Figure FR1: As Fig. 6 but divided by the number of windy conditions.*

**Specific comments**

Line 107: what is the timestep interval?

*The timestep interval is hourly as mentioned in line 98 ("The observational data set includes hourly surface observations").*

Subsection 2.2: Please provide more information on the model run: what is the domain? Boundary conditions? How are wind gusts and cloud cover parameterized in the model? Have they been verified against stations where these data are available?

Boundary conditions: ERA-Interim.

The cloud cover scheme is based on the contributions from grid-scale clouds, sub-grid convective clouds and sub-grid stratiform clouds. The cloud cover scheme is currently based on cloud water and cloud ice only, the contribution from the precipitation categories snow and rain are neglected. (COSMO Documentation Part II, https://www.cosmo-model.org/content/model/cosmo/coreDocumentation/cosmo_physics_6.00.pdf)

Cloud cover is verified with ceilometer measurements (Bollmeyer et al. 2015)

Convection scheme: Tiedtke (1989)

Gusts are estimated following Schulz & Heine (2003; https://www.cosmo-model.org/content/model/documentation/newsLetters/newsLetter03/cnl3-chp3.pdf) and Schulz (2008; www.cosmo-model.org/content/model/documentation/newsLetters/newsLetter08/cnl8_schulz.pdf) with a turbulent and convective gust component.

Line 164: what is the gridding radius of the tracks to produce the "track density"? also in this line it is mentioned "of all cyclones" - it is unclear if the track density is therefore normalized, and if so, what are the units in Fig. 2?

The radius is 750km. The track density is divided by the time period, so Fig. 2 shows the average number per year. We clarified this in the text and the caption.

.

Fig. 2: add lat/lon labels, as these are referred to in the text

Lat/lon labels have been added here and in Fig. A1.

Line 260-261: A bit unclear if 0-10% refer only to NF and CS? And if so, CS in line 261 should be CJ? Is the correlation computed using monthly/seasonal means?

The 0-10% refer to every single feature to the number of stormy time steps. The correlation is computed for both monthly and seasonal means. We clarify this statement.

Fig. 5 and accompanying text: why are the areas east of the Alps masked as white?

We found that this area is affected by the westerly flow, hence, the Alps. The area commonly showed unreasonably high feature probabilities (especially for WJ). As this area is not often affected by extratropical cyclones or rather the associated mesoscale wind features, we decided to mask it. A short explanation will be added.

Line 356: mention how many individual cyclones are composited

We consider 1910 cyclones over around 20,000 timesteps. This will be added to Section 2.4.

Line 355-356: I cannot see the shorter duration of CFC in Figs. 6 and 7. Please clarify.

Fig. 6 and 7 only show when CFC – or any other feature – is common to occur relative to the cyclone lifecycle but do not consider the duration of CFC. As the chosen area does not always include the full development of a cyclone and, hence, does not allow for a meaningful analysis of duration of each feature, we decided not to show a corresponding figure. We added "not shown" to clarify this.

Fig. 8: are the gust data available from observations as well? Have they been verified against observations?

Unfortunately, gust data are not available in the used observational data set, such that a comparison is not possible.

Line 410: Mention also the other excluded regions east of the Alps
This will be done accordingly.

Lines 410-412: it is unclear if the cyclone-relative windy time steps use this relative domain or a 15-degree radius as inferred from line 167. Please clarify
We apologise for the confusion. We always consider the domain +-15° in zonal and -15 to +5° in meridional direction for a consistent analysis.

**Technical corrections**
Thank you for making us aware of these errors. If not noted otherwise, they have been changed accordingly.

Line 157: missing "are" before "targeted"

Line 168, 241: change "consistent to" to "consistent with"

Lines 192, 201, 209, 218: change "percentage points" to "percent" or simply "%"
We would like to clearly separate between '%' and percentage points and keep the section as is.

Line 221: add "of" after "number"

Line 228: replace first "at" with "in"

Line 242: delete "h" from "where"

Line 273: remove "." from "i..e"

Line 343: add "." before "24h"

Line 391: add "relatively cooler" before "land"

Line 425: change "occurrence" to "occurs"

Line 450: replace "larger" with "longer"

---

## Author Response (AR1)

**Review – Suzanne L. Gray**
**Review of 'Identification of high-wind features within extratropical cyclones using a**
**probabilistic random forest - Part 2: Climatology' by Eisenstein et al. submitted to**
**Weather and Climate Dynamics**
**10.5194/wcd-2023-10**

**General comments**:

This is an interesting study describing the results of applying a methodology previously published by the authors to develop a 19-extended-winters climatology of mesoscale wind features in cyclones over Western and Central Europe. The topic is very suitable for this journal. The research is novel in the generation of this type of climatology over this region, and in the method used to obtain the climatology. I have a few questions about the details of the methods (some of the description is confusing, at least to me) and about the robustness of the results. The work would also benefit from a more detailed comparison with previous literature to draw out the novelties of the findings.
The paper is well presented apart from some small language glitches (not all of which I've likely commented on in my review).

Thank you for the thorough review of our manuscript and many valuable comments that help to improve our paper. We have carefully considered each comment. Please find our replies in blue below. Text changes are included in italic and bold when suitable.

**Major specific comments**:

**L105** Here it is stated that the focus of this study is on cyclones occurring in Western and Central Europe. This geographical focus of the study should be included in the paper title.
Thank you for the suggestion. We added "over Europe" at the end of the title.

**Introduction** Given the focus of this paper on the different sources of strong low-level winds, it might be helpful to include a diagram showing where these are located within a classic cyclone as a first figure. The different features are explained in the text but it may be difficult for less knowledgeable readers to visualise these. Related to this comment, some more detail on how the features are defined in probabilistic characterisation would be helpful. I realise that these details are described in the part 1 of this paper but it should be possible for readers to follow part 2 without having first read part 1.
We understand the concern but given the easy and open access of Part 1 we do not think it is necessary to reuse its Figure 1 in Part 2. However, we specifically refer to Figure 1 of Part 1 in the Introduction now and added a few sentences about the distinction of the features in Section 2.3: "*It learned physical consistent characteristics such as decreasing pressure ahead of the cold front, where the WJ is located, and increasing pressure behind the front, where CJ and CS are located. The most important predictor for CFC is precipitation. Furthermore, the warm and cold sectors are indicated by temperature-related parameters. Given the case-to-case variability in the parameters used here, the RF outputs a probability of feature occurrence.*" (l51)

**L73** Here you refer to the Earl et al. study. One interesting result from that study is the different of the prevelence of the features changes depending on whether those generating the top 1% or 0.1% of daily maximum wind gusts are considered. Did you consider changing your windspeed threshold to look at how the relative importance of the features changes?
This is indeed an interesting result to add. We clarified this finding in the introduction: "*Consistent with other studies, they found that WJs and CJs are most common **when looking at the highest 1% of daily maximum wind gust**, while CFC and potential SJs commonly cause **the highest 0.1% of daily maximum gusts**.*" (l85)

Furthermore, we added pie diagrams with higher wind thresholds and a corresponding discussion to Section 3.1 showing an increase of CJ and CFC occurrence with higher thresholds (see response to Section 3.1 below).

**L203** "the proportion of NF is considerably lower in COSMO-REA6 data" – this statement is true but "no feature" strong winds still account for about 30% of the strong winds. Please can you say more about what these strong winds are likely to be caused by and whether they are likely to be linked to cyclones? How often are these NF strong wind regions close to the tracked cyclones (all strong wind regions are within 15 degrees of a cyclone (L167), but are the NF regions relatively more likely to be further away (e.g. beyond 7.5 degrees)?

NF winds are commonly associated with weaker winds (see our response to your next comment and the added Fig. FR2) and, as already mentioned in the preprint, NF detects parts of the CCBa. Looking at a fractional version of the system-relative composite (number of occurrences divided by the number of windy conditions → Figure FR1), the contours of NF shift to the north-eastern quadrant, outside the area of the other features, which show less change. We have included this figure in the Appendix (new Fig. B3).

Furthermore, Part 1 showed that for some cases the spatial independence leads to uncertainty in the area where the introduced features occur, resulting in NF being the most probable feature (see Section 7.4 of Part 1 for more details) or the area between double fronts shows uncertainty (see Section 7.1). This is not commonly the case; however, it might explain the overlap in the southern quadrants to some extent.

It is also evident that the proportion decreases substantially for higher wind thresholds (see next comment).

We added the following to Section 3.1: "*Still, the proportion of NF is accountable for around one quarter to one third of high winds. This is due to several reasons discussed below and in Sect. 3.3 as well as overall higher uncertainty in uncommon cyclone developments, such as double fronts. The reader is referred to Sect. 7 of Part 1 for a detailed discussion (Eisenstein et al., 2022a).*" (l234)

We further added the following to Section 3.3: "*As shown in Appendix B, the relative frequency of the features shows that the NF mostly occurs in the north-eastern quadrant, where other features are rare, and where windy conditions are less common (Fig. B3).*" (l379)

[Figure]

*Figure FR1: As Fig. 6 but divided by the number of windy conditions.*

**Section 3.1** Here the pie charts showing the relative frequency of the different strong wind features are discussed with the pie charts showing percentage occurrence values to 1 decimal

place. It would be useful to have some sense of how robust these relative percentages are to small changes in the datasets used and hence how meaningful the interpretations are. For example, what happens if some of the years are omitted from the datasets used, the definition of strong winds is changed slightly, or the domain of interest (the blue box in Fig. 2) is shifted slightly?

This is a valid point, and we had a further look into this. First, we added pie diagrams for higher wind thresholds to the main text (Figure FR2 → new Figure 4 in the manuscript) showing that the proportion of NF decreases significantly, while especially the proportions of CJ(+SJ) and CFC increase consistently to Earl et al. (2017). Furthermore, three sets of ten randomly chosen winter seasons show only slight changes in proportions (Figure FR3), which is expected given the relatively small fluctuations shown in Fig. 4 (now Fig. 5). A shift in the spatial domain would yield different proportions as can be seen in the Earth-relative plot in the preprint (Fig. 5; now Fig. 6).

We made the following additions to the text: "*To examine how robust these numbers are, we computed three subsets of nine randomly chosen winter seasons. The proportions vary just slightly with an average of around 2% (not shown), as to be expected considering the small fluctuations between winter seasons (see discussion of Fig. 5 below).*

*Figure 4 shows the proportions of each feature for MAXP analogously to Fig. 3m,o (COSMO-REA6 all) but only including grid points where $\tilde{v} \geq 1.2$ (left columns) and $\tilde{v} \geq 1.4$ (right columns). Note that this is only the case for around 1.5% and 0.1%, respectively, of the previously included data points. Considering only $\tilde{v} \geq 1.2$, the proportion of NF reduces by over 75% (Fig. 4a), while it decreases to almost 0% if only grid points with $\tilde{v} \geq 1.4$ are included (Fig. 4c). This suggests that higher winds within the vicinity of a cyclone are mostly associated with one of the introduced features. When NF is neglected, the proportion of CS is reduced by 45% to 58% (Fig. 4b,d) suggesting that while the CS affects a large area, it is less common for it to be the cause of extreme winds. In contrast, the more damaging features CJ and CFC show an increased proportion. CJ shows an increase of about 69% for $\tilde{v} \geq 1.2$ and 111% for $\tilde{v} \geq 1.4$. Consistent to Earl et al. (2017) CFC shows an even stronger increase with over 300% and over 600%, respectively. These results are consistent with the wind characteristics, as will be discussed in Sect. 4. Meanwhile, the proportion of WJ decreases by about 10%, as it usually causes weaker winds compared to the CJ and CFC (e.g., Hewson and Neu, 2015; Earl et al., 2017).*" (l257)

[Figure]

Figure FR2: As Fig. 2 m,o (COSMO-REA6 all; MAXP) but for grid points with (a),(b) $\tilde{v} \geq 1.2$ and (c),(d) $\tilde{v} \geq 1.4$.

[Figure]

Figure FR3: As Fig. 2 m–o (COSMO-REA6 all) but for three different sets of 10 randomly chosen winter seasons.

**L228** For calculation of the seasonal and inter-annual evolution of the wind features, the strong winds need to be present at at least 100 grid points over the domain. The model grid is a latitude-longitude grid with a spacing of 0.0625 degrees. Hence, the size of the grid boxes (in km2) changes with latitude. Is this change taken into account?

The choice of 100 grid points is subjective and was made to remove noise while keeping the rather small-scale CFC feature. This could have been based on an area threshold, too, but given that our data are on a lat-lon grid, we decided to use that. See further discussion below (comment to L271).

**Existing literature** While I don't know of other papers that have produced long term climatologies of mesoscale wind features, there are some other papers that the authors should consider referencing. Rivière et al. (2014, https://rmets.onlinelibrary.wiley.com/doi/10.1002/qj.2412) shows the evolution from a dominant warm to dominant cold conveyor belt jet (see fig. 10 particularly) and Dacre et al. (2012, https://journals.ametsoc.org/view/journals/bams/93/10/bams-d-11-00164.1.xml) describes the generation of a cyclone "atlas" which includes analysis of the warm and cold conveyor belts. I also suggest attempting a more quantitative comparison with the existing literature that is cited. The term "mostly consistent" is used in the conclusions – can you provide a deeper comparison? How do the cyclones over the Western and Central Europe region analysed here compare with those over the North Atlantic or UK? It would also be helpful to contrast the descriptions of the cold and warm jets here with some much older literature. For example, for me the classic paper on cyclone structure is Browning and Roberts (1994, https://rmets.onlinelibrary.wiley.com/doi/epdf/10.1002/qj.49712052006). See also Browning (2005, https://rmets.onlinelibrary.wiley.com/doi/10.1256/qj.03.201). This deeper analysis would help the reader understand what new knowledge is generated by this research. Currently this seems rather modest as it is detailed in just two sentences in the conclusions as follows: "The large number of storms investigated helped revealing the large variability in the location of CFC in a system-relative framework. Other differences to the literature include the

time of occurrence of the CJ already several hours before the time of maximum depth in contrast to Hewson and Neu (2015)."

Thank you for the recommendation. We included the suggested papers Dacre et al. and Riviere et al. in the introduction. We also considered the other recommendations, however, we believe that their inclusion is not critical within the scope of our paper, where we are interested in the occurrence of the introduced wind features rather than a historical overview.

We further included more existing literature in the result chapters and conclusions, comparing findings and clarifying the added knowledge through this climatology.

Additions to the introduction:

"*Dacre et al. (2012) describe the development and compilation of an extratropical cyclone atlas using 200 extreme North Atlantic cyclones over a 20-year period. The atlas includes composites of horizontal and vertical cyclone structure, multiple parameters (e.g., cloud cover, wind and relative humidity) and cyclone evolution, while also identifying the warm and cold conveyor belts and dry intrusions.*" (l63)

"*A study by Riviere et al. (2015) suggests that in early stages a cyclone is dominated by the WJ and later by the CJ.*" (l79)

Additions to the conclusions:

"*While previous literature suggests a cyclone to be first dominated by the WJ and second by the CJ (e.g., Hewson and Neu, 2015; Riviere et al. (2015), this climatology further revealed the occurrence of CFC in early development stages and the dominance of CS in later ones.*" (l494)

"*The large number of storms investigated helped to reveal the large variability in the location of CFC in a system-relative framework, **similar to the blurring of frontal boundaries in composites discussed in Dacre et al. (2012).***" (l498)

**Minor specific comments**:

**L2** I suggest replacing "high winds" with "strong winds" as used on the previous line as "high" could refer to the altitude of the winds rather than their strength. Please also consider other places where "high" or "highest" is used.

While we understand this concern, we think the usage of "high" synonymously to "strong" is clear in the paper. We change this in some places but decided to not change this throughout the paper to be consistent to Part 1 (and the title).

**L3** "strong cold-sector winds" are not a "dynamical feature" (unlike the other mesoscale features considered).

Thank you for drawing attention to this. We remove "dynamical" here.

**L35** Here it is stated that cold frontal convection is more common in cyclones following the Norwegian cyclone model. Can you point to evidence supporting this statement? Although cyclones following the Shapiro-Keyser conceptual model undergo frontal fracture, the cold front can be stronger than the warm front in idealised lifecycle simulations of these cyclones with the converse true for Norwegian-type cyclones (e.g., see Fig. 3 of Shapiro et al. 1999: https://link.springer.com/chapter/10.1007/978-1-935704-09-6 14).

Several Shapiro-Keyser cyclones in the recent years showed a less active cold front (e.g., Egon 2017, Xavier 2017, Friederike 2018). One paper suggesting that the warm front is stronger than the cold front in Shapiro-Keyser cyclones (in contrast to Norwegian cyclones, which is suggested to have a strong cold front) is Catto (2016; https://onlinelibrary.wiley.com/doi/10.1002/2016RG000519, e.g. Fig. 14).

We realise that we cannot provide more statistical evidence such that we rephrased this statement to clarify this:

"***In Shapiro-Keyser cyclones (Shapiro and Keyser, 1990) the warm – or bent back – front is usually the stronger front (Catto, 2016). Considering recent cases of Shapiro-Keyser cyclones (e.g., Egon 2017, Xavier 2017, Friederike 2018; see Eisenstein et al., 2022a),** this feature **appears to be** more common following the Norwegian cyclone model (Bjerknes, 1919).*" (l36)

**L55** Regarding tracking cyclones over the Mediterranean, you might be interested in this recent paper: https://wcd.copernicus.org/preprints/wcd-2022-63/
Thank you for the recommendation.

**L59** Probably worth pointing out that the calculation of Laurila et al. considers the wind extremes using monthly values (i.e., the extreme wind factor is the monthly 98th percentile/monthly mean)
Thank you for pointing this out. We clarified this in the text: "*They focus on the North Atlantic and Europe by defining an extreme wind factor, which is the **monthly** 98th percentile divided by the **monthly** mean wind speed.*" (l67)

**L66** In this paragraph you should point out that the studies referred to consider different regions e.g., the Parton et al. and Earl et al. studies only consider the UK and so may not be representative of cyclones in general.
We made the following text changes: "*According to them [Parton et al.], warm-sector events are the most common cause of strong winds with around 40%, while cold-frontal events make up around 24% **over the investigated area, which may not be representative of cyclones in general.***" (l77)
"*However, their [Earl et al.] focus lays solely on winter storms over the UK similar to Parton et al. (2010).*" (l86)

**L107** Here it states that "January 2001 to December 2019" data is used whereas at the start of this section the observational dataset is described as extending to mid-2020.
Although the data set is available until mid-2020, we do not include 2020 to get the same number (19) for each month and being as close to the COSMO-REA6 data set as possible. We clarify this in the text by adding: "***For the climatology,** we include all time steps from January 2001 to December 2019, i.e., a total of 114 months.*" (l119) and further in Section 2.2 (see next comment).

**L114** "we concentrate here on October 2000 to March 2019, i.e., a minor shift of three months". The model data starts 3 months earlier than the observational data but ends 9 months earlier. The reason this doesn't make any difference is because only extended winter data is considered - this could be made clearer.
We apologise for the confusion. The COSMO-REA6 data set is available from 1995 to mid-2019, while the observations are available from 2001 to mid-2020 (see previous comment). To make the data sets as comparable as possible, we include 19 extended winter seasons with the same number of months, however, a slight shift of three months.
We clarified this in Section 2.2: "*For a fair comparison of the two data sets, the chosen time period for the climatology is as close as possible while including 19 extended winter seasons each. This means, while the observations cover January 2001 to December 2019, the COSMO-REA6 is used for October 2000 to March 2019, i.e., with a minor shift of three months.*" (l125)

**L114** Can you please state the number of data points used to calculate the 98th percentile at each location and date. I think it's 210 (21 days of data for each of 10 years) but it would be good to know if I'm correct. The 98th percentile winds at any grid point will be strongly dependent on whether a localised strong wind region happens to cross that grid point on one of these 210 days. How does the value of this 98th percentile (or the lower "windy" threshold) vary spatially for a sample time snapshot? Also, is it correct that the wind speed threshold at a given time and place will be different in absolute terms for the model and observational data because the normalisation is calculated separately for the model and observational data?
210 data points is correct for COSMO-REA6. For Part 1 we tested the computation with +-30 days for an exemplary month but could not see substantial differences. To save computational costs we decided to keep +-10 days instead.

Figure FR4 shows a snapshot for COSMO-REA6 (left) and overlayed with observations (right). While only including 10 years – instead of 20 as done for the observational data set – might give exceptionally strong events more weight, the comparison of COSMO-REA6 to observations shows only slight differences, which are probably due to the different data sets instead of single events. The biggest differences occur at mountain peaks as to be expected.

[Figure]

*Figure FR4:* 98th percentile of wind speed in m/s for 01 January, 0 UTC. COSMO-REA6 (left) and COSMO-REA6 overlayed with observations (right).

**L129** Here it states "Here, we apply RAMEFI to station observations and COSMO-REA6 data under windy conditions during the extended winter months, regardless of whether a storm occurred or not. However, we later filter the output for cyclone occurrence as discussed in Sect. 2.4." Please can you clarify if any of the results shown are not filtered for cyclone occurrence? I wondered if the filtering for cyclones might only be applied in Sections 3.2 and 3.3 or even just in Section 3.3.
We apologise for the confusion. All results are filtered by cyclone and area to be consistent throughout the paper. We clarified this throughout the text and included the following sentence in Section 2.4: "*Overall, these filters lead to 1910 cyclones over around 20,000 time steps, which are included in the analysis.*" (l200)

**Fig. 2** Can you provide more details on how the track density is calculated? This field is very smooth in the plot so I suspect that there is some averaging occurring (typically track densities are plotted as the number of tracks within a certain area, such as a 5 degree spherical cap, of each grid point). I don't think it's possible that there are up to about 3 storms with their MSLP centre located at a single grid point each winter. Also, it would be more helpful to readers for the track density per year to be plotted rather than the number over the 19 years.
The track density calculation considers a radius of 750km. We calculate the sum of previously identified cyclone tracks in any grid points that are at least once within the domain. The track density is then the total accumulated number of tracks divided by the given time period to get the track density per year.
We clarified this in the text and the caption: "*The algorithm primarily searches for the minimum p in the vicinity of a $\nabla^2 p$ maximum as cyclone centres **within a radius of 750km.***" (l183)
Caption: "***Cyclone** track density **in number of** cyclones **per year per (° lat)$^2$** within the examined time period (2000--2019). The blue box represents the study area, **with hatching indicating excluded areas***."

**Fig. 3** The domain used for analysis of the observational data is, I think, smaller than that used for the model data (see L105 and L118) with the latter extending further east and north. However, a further domain is specified in L169. If these domains are different then how does this difference affect the comparison between the results from the observations and model output shown in this figure? What would the figure look like if the same domains were used?
First of all, we found a mistake in the manuscript. The analysed zonal extent for COSMO-REA6 is 10°W to 25°E (not 30°E). Figure FR5 shows pie diagrams for observations (top row)

and COSMO-REA6 land grid points (lower row) for the same domain (10°W to 20°E, 45°N to 60°N) showing just slight differences to the original pie diagrams in Fig. 3.

[Figure]

*Figure FR5: As Fig 2a—h for observations (top row) and COSMO-REA6 grid points over land (bottom row) for a consistent area of 10°W to 20°E, 45°N to 60°N. The first and third column show MAXP, while the second and fourth column show ACCP.*

**Fig. 4** Does this plot consider a feature as occurring if it has at least 100 grid points with any percentage of likelihood at each grid point or only if it is the most probable feature at a given grid point? What is the definition of a stormy time step (is it that any feature is present at at least 100 grid points?).
It is considered when it is the most probable feature. We added this for clarification: "*This is computed as the sum of stormy time steps during which a certain feature is detected as **the most probable feature** at at least 100 grid points […].*" (l272)
The definition of stormy time steps is written in l. 171f but has been clarified to: "*Time steps with a cyclone moving through and at least 5% **of the area showing** windy conditions are referred to as 'stormy time steps'.*" (l199)

**L242** "However, the recent two decades w[h]ere characterised by an unusual number of storms in October," - please provide evidence for this statement.
This statement arose from a submitted paper (Mömken et al. 2023). Looking at insurance data and SSI (at least over Germany), we found almost no noteworthy winter storms in November in our time period while we found several in October (e.g., Christian/St. Jude's Day Storm 2013, Xavier 2017, Herwart 2017). We rephrased this statement: "*However, the recent two decades **show a larger number of noteworthy** storms in October **(e.g., Christian 2013, Xavier 2017, Herwart 2017) compared to November**, […].*" (l287)

**L243** I'm confused by the statement "given the lower threshold of ˜v - leads to a larger number of stormy time steps in the 19 years". ˜v is the wind speed normalised by it's 98th percentile value on the day of the year (and time of day and location) over the period 2005-2015 and stormy timesteps are those in which the windspeed exceeds 80% of ˜v. Given this normalisation I'd expect the number of stormy days not to vary too much between months (as that is what the normalisation is trying to achieve). I suspect I've misunderstood something here - is it that the location specific nature of the windspeed threshold simply means that it picks out all stormy events? Please can you explain?
We apologise for the confusion; the statement indeed was not phrased well. Given the normalisation, the absolute wind speed can be slightly lower in October compared to

November for the grid point to be considered "windy" due to the slightly lower $v_{98}$. While the main reason for the higher number in October is the one discussed before, this slightly lower $v_{98}$ in October might further lead to more stormy time steps.

We clarified this in the text: "*However, the recent two decades **show a larger number of noteworthy** storms in October **(e.g., Christian 2013, Xavier 2017, Herwart 2017) compared to November**, which leads to a larger number of stormy time steps in the 19 years. **This difference might be further enhanced by the slightly lower $v_{98}$ in October compared to November**.*" (l287)

**L247** Here the frequency of the cold jet is linked to that of cold air outbreaks. Please can you explain why these two phenomena could be expected to be linked?

The two phenomena are not directly linked synoptically, and our statement is indeed misleading. One reason why cold air outbreaks lead to a higher frequency of detected CJ might be that the normalised potential temperature is likely to show negative values despite the normalisation, which is connected with a higher probability of winds being allocated to CJ or CS winds (see Part 1, Fig. 10). However, as it is not a very important predictor (see Part 1, Fig. 8 and 9), the effects to the detection overall are usually small.

We clarified this: "*Recall that the $\tilde{\theta}$ predictor is a normalised parameter, such that a cooling Arctic with progressing winter and possibly more cold-air outbreaks, **i.e., lower values of $\tilde{\theta}$, might lead to higher numbers of high winds being allocated to the CJ – and CS**.*" (l294)

**L271** The definition of "windy" is different here to that in L167. L167 defines it as strong winds "in the vicinity of 15 degrees of the cyclone centre," which implies a circular region around the cyclone whereas here a box with ±15 degrees in the zonal direction and -15 to 5 degrees in meridional direction is stated. Which is correct? Also, how is the reducing zonal extent (in km) with latitude accounted for (e.g., 30 degrees in the zonal direction varies from 2330 to 1390 km between 45 and 65 degrees North)?

We apologise for the confusion. For the whole climatology, we considered ±15° in the zonal direction and -15 to 5° in meridional direction as corrected in Section 2.4: "*Here, all grid points showing windy conditions in the vicinity of **15° in zonal direction and -15° to 5° in meridional direction** of the cyclone centre […].*" (l191)

As mentioned above, the changing distance in km for different latitudes was not considered, as our data is on a lat-lon grid. This is similar to other papers using cyclone-relative analysis (e.g., Sinclair et al., 2020 (https://wcd.copernicus.org/articles/1/1/2020/), Priestley et al., 2022 (https://wcd.copernicus.org/articles/3/337/2022/)). We do not anticipate that a change to area-thresholds would change the overall statements in our discussion. It would be desirable to test this effect, but the computational costs to do this for the whole climatology are quite high.

We add a note on this in the text: "*Note that the changing zonal extent with latitude is neglected in this analysis. However, we do not expect this to have a significant impact on our main conclusions.*" (l196)

**299** Here it is stated "As orography can induce convection, CFC might be detected without the occurrence of a cold front or even cyclone". I thought only windy features were identified close to a cyclone were considered though (see above comment and also the comment relating to line 129).

This is indeed misleading. What we wanted to say is that wind speeds not associated with one of the features but for example associated with a strong pressure gradient in the vicinity of a cyclone and hitting orography can induce convection and be identified as CFC. For example, this commonly happens over the Norwegian mountains.

We changed this statement to: "*As orography can induce convection, CFC might be detected without the occurrence of a cold front, **but where high wind speeds are associated with a strong pressure gradient or other features combined with orographic convection**.*" (l346)

**Fig. 6** I think a constant latitude has been assumed here (of 50 degrees) in the calculation of the distance circles as they appear circular. If the variation of zonal distance with latitude had

been considered the distance circles would not be circular. This assumption should be stated in the caption. Please also add to the caption that the grey shading in (a) is for the NF winds. Yes, this is correct. The caption has been modified accordingly: "*Occurrence of the identified mesoscale wind features (a) relative to the cyclone centre **--including NF in grey--** and (b) relative to the cyclone life cycle. The contours in (a) show the area with most feature occurrences, including 25% (filled contours) and 50% (outer contours) of the detected features. Black circles show the distance to the cyclone centre in 250km increments **using 50°N as a reference latitude**.*"

**Fig. 6a** How is the location of the NF winds close to the cyclone centre consistent with Fig. 5b which shows NF winds are typically found to the south and north of the main storm track? The histograms in Fig. 5 are divided by the number of windy conditions. This is not done in Fig. 6a. Doing so leads to NF mostly occurring in the north-eastern quadrant as discussed above (see Fig. FR1).

**L373** "RR values over 1m s−1". "RR" isn't defined in this paper but based on the earlier paper by the authors I guessed that it stood for precipitation amount. However, the units are then incorrect. Please define RR. Precipitation amount *RR* has been defined in Section 2.1 (formerly line 99, now line 111). However, we apologise for the confusion regarding the units. These are of course mm h$^{-1}$.

**Fig. 8** What heights are all the fields considered at? We added the height in Section 2 (Data and methods): "*mean sea level pressure (p), **2m** air temperature (T), wind speed **at 10m** (v), wind direction **at 10m** (d) and precipitation amount (RR).*" (l110)

**L383** Here it is stated that it is not surprising that the highest gust ratio is found for the CFC features because convection is associated with high instability and turbulence. The grid spacing of the model is about 6 km. Does it use a convection scheme or is convection represented explicitly? How are the gusts diagnosed in the model? Related to this point, it would be interesting to know how the climatology of model gusts shown in Section 4 compares to a similar climatology generated from the observations given the likely parametrization dependence of the model-derived gusts. Have you considered this? The model uses the mass flux scheme from Tiedtke (1989) for subgrid-scale convection. Following a comment of Reviewer 2, we included further parametrisation details in Section 2.2: "*The data set covers the European CORDEX (Coordinated Downscaling Experiment) domain with a grid spacing of 0.055°, i.e., roughly 6km, and **uses ERA-Interim data (Dee et al., 2011) as boundary conditions**. […] **The model uses a convection parametrisation by Tiedtke (1989) and wind gusts are estimated following Schulz and Heine (2003) and Schulz (2008) with a turbulent and a convective gust component. The cloud cover is based on cloud water and cloud ice and considers grid-scale, sub-grid convective and sub-grid stratiform clouds (Doms et al., 2021) and verified using ceilometer data (Bollmeyer et al., 2015)**. For a detailed **description of the data set**, we refer to Bollmeyer et al. (2015) **and Doms et al. (2021).**" (l123) Unfortunately, the observational data set we use does not include wind gusts, such that a comparison is not possible.

**L425** What is the evidence from your analysis that the CFC feature requires a convective trigger? The term "trigger" is also vague: do you include both forced ascent and ascent due to the release of convective instability? This statement indeed included poor choices of words. Of course, CFC is triggered by the vertical movements along the cold front itself. However, daytime heating, frictional convergence and orography can contribute to enhance the CFC. We rephrased this point to: "*Occurs almost exclusively over land, **where, e.g., daytime heating and frictional convergence, can strongly enhance the development of convection along the cold front, and is also***

*detected particularly around mountainous **areas** (sometimes orographic triggering independent of cold front)*" (l474)

**Technical errors:**
Thank you for making us aware of these errors. All have been changed accordingly.

L34 "causing" → "it causes".

L44 "As CJ" → "As the CJ"

L51 "perception on" → "perception of"

L74 "convection-induces" → "convection-induced"

L85 "northeastern" → "northeast"

L106, L170, L190 "less than" → "fewer than" (n.b. you should use "fewer than" if you can count the objects e.g., fewer bottles of lemonade but less lemonade). Please also check elsewhere in the paper.

L110 "Ger- man" → "German"

L156 Remove full stop before start of 1st sentence.

L168 "in altitudes above 800m consistent to Part 1" → "at altitudes above 800m consistent with Part 1"

L197 "perscpective" - spelling.

L232 "any" → "every"

L239 "amount" → "number" (because the number of stormy timesteps can be counted).

L241 "consistent to Feser et al." → "consistent with Feser et al."

L242 "where" → "were"

L244 "corresponds" → "correspond"

L301 "of CJ" → "of the"

L306 add "this" before "suggests"

L326 "(Gentile and Gray, 2023)" → "Gentile and Gray (2023)".

L330 "particular" → "particularly"

L446 "helped revealing" → "helped to reveal"

L450 "larger" → "longer"

**Reviewer 2**
**'Review of "Identification of high-wind features within extratropical cyclones using a probabilistic random forest – Part 2: Climatology" by Lea Eisenstein et al.'**
**10.5194/wcd-2023-10**

This paper provides a valuable 19-year, extended-winter climatology of strong winds related to cyclones in northwestern Europe. The novel aspect is the use of the RF-based approach recently published by the authors to label the near-surface winds according to distinct mesoscale features that produce the strong winds: the warm jet, cold jet, cold-front convection and cold-sector winds, along with a 'no feature' category. This distinction indeed shows different Earth-relative and cyclone-relative characteristics, and it opens the door to understanding the predictability of cyclone-induced winds which would likely depend on the feature.

The scope fits WCD well, the text is very well written and the figures compelling. I enjoyed reading the manuscript and I overall support its progress towards publication. There are, however, some aspects needing attention and clarification before its acceptance.

Thank you for the thorough review of our manuscript and many valuable comments that help to improve our paper. We have carefully considered each comment. Please find our replies in blue below. Text changes are included in italic and bold when suitable.

**Major comments**
1. Some key aspects related to the RAMEFI method need to be enhanced in the manuscript, so that Part 2 can stand alone: please explain how can station data from only 12 cyclones construct a sufficient training dataset for its applicability to (i) all cyclones in the climatology, to (ii) other, gridded data, and to (iii) location-specific features, such as land/sea differences or orographic effects.
   We made several additions regarding the following aspects:
   (i) While 12 case studies seem to be quite few, they represent a healthy diversity of cyclone developments and features including very intense and more moderate cyclones with differing storm tracks. Overall, almost 280 time steps leading to almost 80,000 data points are included in training the RF. The promising evaluation using a cross-validation approach in Part 1 suggests a reliable detection of the features in long-term data for most cyclones: "*These case studies are picked to capture a healthy diversity of cyclone developments and features, i.e., includes very intense and more moderate cyclones with differing storm tracks. Given this diversity and the promising evaluation of the method in Part 1, we assume a reliable detection of the features in long-term data for most cyclones.*" (l141)
   (ii) We clarified this in Sect. 2.3: "*Further, the statistical evaluation **of the application on COSMO-REA6 data** in Part 1 demonstrates that RAMEFI generates reliable identifications for **gridded data** despite being trained on surface observations.*" (l148)
   (iii) While the normalisation of *v* and *θ* brings some independence of location-specific effects and we remove grid points/stations over 800m and the Balkans, some other areas are still affected by nearby orography (e.g., Norway) and results should be treated with some caution here. As written in the manuscript (Sect. 2.3 and Appendix A), we did not statistically evaluate RAMEFI over the ocean and smaller differences are to be expected, but looking at several case studies did not show any obvious issues.

2. It is not entirely clear how the distinction is made among the WJ/CJ/CS/CFC during the initial subjective labelling in the vicinity of the front. Since the front is a narrow and slanted surface, the location of the front line changes with height, and accordingly the attribution of the windy grid points on both its sides. It is unclear how the station-based identification of convection relates to the commonly-used front detection by the 850-hPa temperature/wind/humidity gradients and why these features are mostly absent from oceanic locations. In this context I find the comment in lines 299-300 somewhat

disturbing. How should one distinguish between winds caused by frontal convection from any other isolated convective downdrafts? It would be beneficial to clarify these points, so that the high uncertainty of the CFC winds can be reduced or better understood.

Please note that we do not identify the front itself, but the convection induced in association with a front. While the most important predictor is *RR*, further predictors ensure the location in the frontal region, such that post-CFC is identified as CS rather than CFC. These include for example Δ*p* and *θ*. However, in orographic regions such as the Norwegian mountains, CFC can be falsely detected due to orographically induced convection.

We included the following sentences in the introduction: "*It learned physical consistent characteristics such as decreasing pressure ahead of the cold front, where the WJ is located, and increasing pressure behind the front, where CJ and CS are located. The most important predictor for CFC is precipitation. Furthermore, the warm and cold sectors are indicated by temperature-related parameters. Given the case-to-case variability in the parameters used here, the RF outputs a probability of feature occurrence.*" (l51)

And rephrased a sentence in Sect. 3.2: "*As orography can induce convection, CFC might be detected without the occurrence of a cold front, **but where high wind speeds are associated with a strong pressure gradient or other features combined with orographic convection**.*" (l346)

3. The description of the application of the RAMEFI method to both station and model data should be enhanced. It was not clear to me from the text whether the method is carried out independently for each data set, and therefore providing independent probabilities for each, and why is it designed so? Therefore I wasn't sure if the differences among the datasets seen in Fig. 3 stem from the fact that there are stations only over land, and they are not evenly distributed, or whether there are additional biases from the application of the method to each dataset separately?

RAMEFI makes decisions based only on a single station or grid point at a certain time. Hence, it is not just applied to the data sets individually but to each data point, where one data point represents one station/grid point at a certain time. We added the following sentence for clarification: "*RAMEFI is spatially independent, such that the output probability is computed individually for each station or grid point.*" (l151)

Small biases are to be expected as the evaluation of COSMO-REA6 is slightly worse than for the observation data set, it was trained on (see Sect. 6 of Part 1). This is why we compare the two data sets in the scope of this section. More substantial differences are rather due to data point distribution, missing data in the observations and overall differences in the development of a cyclone in the data sets.

4. The fact that NF is not randomly distributed in Figs. 6 and 7 suggests that some wind is currently associated to NF even though it should be coherently linked to the cyclone and is therefore mislabeled. Please quantify the sensitivity of the results to the labelling procedure, and estimate how much of the NF signal in Figs. 3 or 5 is found in the vicinity of cyclones.

Fig. 6 and 7 show the absolute occurrence, i.e., independently of the other features or number of windy data points. The south-western area of a cyclone is characterised by higher wind speeds, while the north-eastern quadrant shows windy conditions less often. The relative occurrence shows the NF feature mostly in the north-eastern region, as shown in Fig. FR1, which has been added to the Appendix (Fig. B3).

This climatology showed us that for example the CCBa, i.e., the CJ before it wraps around the cyclone centre, is more common than we expected when developing the method as a distinct peak of NF occurrence is visible there. Although, the CCBa could be considered as an additional feature in future work, the focus here is on WJ, CJ, CS and CFC, which are the most common causes of high-winds, and their detection is not hindered by CCBa being included in NF.

The absolute occurrence of NF in the south-western quadrant is mostly caused by uncertainty, e.g., due to double fronts or untypical structures (see Sections 7.1 and 7.3 in Part 1).

We added the following discussion on this:

Section 3.1: "*Still, the proportion of NF is accountable for around one quarter to one third of high winds. This is due to several reasons discussed below and in Sect. 3.3 as well as overall higher uncertainty in uncommon cyclone developments, such as double fronts. The reader is referred to Sect. 7 of Part 1 for a detailed discussion (Eisenstein et al., 2022a).*" (l234)

Section 3.3: "*As shown in Appendix B, the relative frequency of the features shows that the NF mostly occurs in the north-eastern quadrant, where other features are rare, and where windy conditions are less common (Fig. B3).*" (l379)

[Figure]

*Figure FR6: As Fig. 6 but divided by the number of windy conditions.*

**Specific comments**

Line 107: what is the timestep interval?

The timestep interval is hourly as mentioned in line 98 – now line 110 ("*The observational data set includes hourly surface observations*").

Subsection 2.2: Please provide more information on the model run: what is the domain? Boundary conditions? How are wind gusts and cloud cover parameterized in the model? Have they been verified against stations where these data are available?

Boundary conditions: ERA-Interim.

The cloud cover scheme is based on the contributions from grid-scale clouds, sub-grid convective clouds and sub-grid stratiform clouds. The cloud cover scheme is currently based on cloud water and cloud ice only, the contribution from the precipitation categories snow and rain are neglected. (COSMO Documentation Part II, https://www.cosmo-model.org/content/model/cosmo/coreDocumentation/cosmo_physics_6.00.pdf)

Cloud cover is verified with ceilometer measurements (Bollmeyer et al. 2015)

Convection scheme: Tiedtke (1989)

Gusts are estimated following Schulz & Heine (2003; https://www.cosmo-model.org/content/model/documentation/newsLetters/newsLetter03/cnl3-chp3.pdf) and Schulz (2008; www.cosmomodel.org/content/model/documentation/newsLetters/newsLetter08/cnl8_schulz.pdf) with a turbulent and convective gust component.

We added the following information: "*The data set covers the European CORDEX (Coordinated Downscaling Experiment) domain with a grid spacing of 0.055°, i.e., roughly 6km, and **uses ERA-Interim data (Dee et al., 2011) as boundary conditions**. […] **The model uses a convection parametrisation by Tiedtke (1989) and wind gusts are estimated following Schulz and Heine (2003) and Schulz (2008) with a turbulent and a convective gust component. The cloud cover is based on cloud water and cloud ice and considers grid-scale, sub-grid convective and sub-grid stratiform clouds (Doms et al., 2021) and verified using ceilometer data (Bollmeyer et al., 2015)**. For a detailed **description of the data set**, we refer to Bollmeyer et al. (2015) **and Doms et al. (2021).**" (l123)

Line 164: what is the gridding radius of the tracks to produce the "track density"? also in this line it is mentioned "of all cyclones" - it is unclear if the track density is therefore normalized, and if so, what are the units in Fig. 2?

The radius is 750km. The track density is divided by the time period, so Fig. 2 shows the average number per year. We clarified this in the text and the caption: "*The algorithm primarily searches for the minimum p in the vicinity of a $\nabla^2 p$ maximum as cyclone centres **within a radius of 750km.**" (l183)

Caption: "**Cyclone** track density **in number of** cyclones **per year per (° lat)$^2$** within the examined time period (2000--2019). The blue box represents the study area, **with hatching indicating excluded areas**."

Fig. 2: add lat/lon labels, as these are referred to in the text
Lat/lon labels have been added here and in Fig. B1.

Line 260-261: A bit unclear if 0-10% refer only to NF and CS? And if so, CS in line 261 should be CJ? Is the correlation computed using monthly/seasonal means?

The 0-10% refer to every single feature to the number of stormy time steps. The correlation is computed for both monthly and seasonal means. We clarified our statement as follows: "*Overall, **all** features have no to very weak positive correlation with the number of stormy time steps (0-10%).*" (l307)

Fig. 5 and accompanying text: why are the areas east of the Alps masked as white?

We found that this area is affected by the westerly flow, hence, the Alps. The area commonly showed unreasonably high feature probabilities (especially for WJ). As this area is not often affected by extratropical cyclones or rather the associated mesoscale wind features, we decided to mask it (see also Section 6.1 of Part 1).

A short explanation has been added in Section 2.4.: "*Furthermore, following Part 1 we exclude the Balkans, where topography is complex and where winter storms are rare. For example, in mountainous regions, the Foehn effect might be falsely identified as WJ. Hence, such areas are removed from the climatology, as indicated by the hatching in Fig. 2.*" (l193)

We further included a note in the caption: "*Hatching indicates grid points with an altitude above 800m **(dots) and excluded areas east of the Alps (lines)**.*"

Line 356: mention how many individual cyclones are composited
We added the following sentence to Section 2.4: "*Overall, these filters lead to 1910 cyclones over around 20,000 timesteps, which are included in the analysis.*" (l200)

Line 355-356: I cannot see the shorter duration of CFC in Figs. 6 and 7. Please clarify.
Fig. 6 and 7 only show when CFC are commonly occurring relative to the cyclone lifecycle but does not consider the duration of CFC. As the chosen area does not allow for a meaningful analysis of duration, we decided to not show a corresponding figure. We added "(*not shown)*" to clarify this.

Fig. 8: are the gust data available from observations as well? Have they been verified against observations?
Unfortunately, gust data is only available over Germany, such that we did not include it in our analysis.

Line 410: Mention also the other excluded regions east of the Alps
This has been done accordingly: "*The considered area includes Western and Central Europe, however, excluding grid points above 800m altitude **and the Balkans**.*" (l459)

Lines 410-412: it is unclear if the cyclone-relative windy time steps use this relative domain or a 15-degree radius as inferred from line 167. Please clarify
We always consider the domain +-15° in zonal and -15 to +5° in meridional direction for a consistent analysis. This has been corrected in Section 2.4.: "*Here, all grid points showing windy conditions in the vicinity of **15° in zonal direction and -15° to 5° in meridional direction** of the cyclone centre […].*" (l191)

**Technical corrections**
Thank you for making us aware of these errors. If not noted otherwise, they have been changed accordingly.

Line 157: missing "are" before "targeted"

Line 168, 241: change "consistent to" to "consistent with"

Lines 192, 201, 209, 218: change "percentage points" to "percent" or simply "%"
We would like to clearly separate between '%' and percentage points and kept the section as is.

Line 221: add "of" after "number"

Line 228: replace first "at" with "in"

Line 242: delete "h" from "where"

Line 273: remove "." from "i..e"

Line 343: add "." before "24h"

Line 391: add "relatively cooler" before "land"

Line 425: change "occurrence" to "occurs"

Line 450: replace "larger" with "longer"

---

## Referee Report (RR1)

**Review of 'Identification of high-wind features within extratropical cyclones using a probabilistic random forest - Part 2: Climatology over Europe' by Eisenstein et al. submitted to Weather and Climate Dynamics**

**General comments:**

This is my second review of this interesting study describing the results of applying a methodology previously published by the authors to develop a 19-extended-winters climatology of mesoscale wind features in cyclones over Western and Central Europe. The authors have responded to my earlier questions and concerns very thoroughly, including making appropriate edits to the paper. I have no further concerns relating to my original review. In reading the revised paper I spotted a few rather minor language edits and science clarifications that should be considered and these are listed below. I recommend that the paper is accepted for publication after these comments have been considered and I look forward to seeing it in Weather and Climate Dynamics.

**Minor specific comments:**

**Rebuttal letter** On p4 of your rebuttal letter you examine the robustness of your findings by, amongst other things, considering subsets of winter seasons. In the description in the rebuttal letter and caption of the associated figure, FR3, you say that 10 randomly chosen seasons were considered, however in the text added to the paper (which is also repeated in the rebuttal letter) you say that nine seasons were considered. Which is it?

**L38** I suggest changing "this feature" to "the CFC feature" for clarity (in my first reading of this sentence I thought "feature" refered to the bent-back front mentioned in the previous sentence).

**L60** By "features" here do you mean the storm tracks over the North Atlantic and North Pacific? This could be written more clearly.

**Section 2.1** It would be useful add the source of the observational data set here, as stating that the observations are only available over land (which I think is true). The source is stated in the data availability section of both parts of the paper but it would be helpful to repeat it here as well.

**L136** I appreciate that you showed in your rebuttal letter figure FR5, a figure that used the same domain for the COSMO-REA6 data as for the observational data, for comparison with Fig. 3a–h in the paper. As you still use different domains for these two datasets in the figures shown in the paper it would be helpful to note this in the paper and say briefly why you chose to use the different rather than consistent domains.

**Section 2.4** It would be useful to add here if there is a constraint on the minimum length (in time or space) of the cyclone tracks.

**p10** Consider combining some of the 3 short paragraphs near the end of this page.

**L374 & 401** Gentile and Gray (2023) didn't introduce the term CCBa, it was used previously by Earl et al. (2017). I don't know whether Earl et al. were the first to use it though.

**Technical errors:**

**L95** "cause" → "have" (otherwise you are saying that winds cause winds!).

**L129** I would say "at 10m" etc. (rather than "in 10m"); this applies three times in this sentence.

**L163** "characteristics in other..." → "characteristics of other...".

**L164** "different" → "differently".

**L243 and elsewhere** To be horribly pedantic, "less" should be "fewer" here (if you can count the items you should use "fewer" whereas you would use "less" for an amount, e.g., less time).

**L333** "they might" → "it might" (because frequency is singular) or say "as WJs might"

**L429** What does "they" refer to here? Strong winds?

---

## Author Response (AR2)

**Review of 'Identification of high-wind features within extratropical cyclones using a probabilistic random forest - Part 2: Climatology over Europe' by Eisenstein et al. submitted to Weather and Climate Dynamics**

We would like to thank the reviewers and editor again for their review of our manuscript.
Below we addressed the remaining comments of Reviewer 1 in blue. Text changes are included in italics when suitable. Line numbers correspond to the revised manuscript.

**General comments:**

This is my second review of this interesting study describing the results of applying a methodology previously published by the authors to develop a 19-extended-winters climatology of mesoscale wind features in cyclones over Western and Central Europe. The authors have responded to my earlier questions and concerns very thoroughly, including making appropriate edits to the paper. I have no further concerns relating to my original review. In reading the revised paper I spotted a few rather minor language edits and science clarifications that should be considered and these are listed below. I recommend that the paper is accepted for publication after these comments have been considered and I look forward to seeing it in Weather and Climate Dynamics.

**Minor specific comments:**

**Rebuttal letter** On p4 of your rebuttal letter you examine the robustness of your findings by, amongst other things, considering subsets of winter seasons. In the description in the rebuttal letter and caption of the associated figure, FR3, you say that 10 randomly chosen seasons were considered, however in the text added to the paper (which is also repeated in the rebuttal letter) you say that nine seasons were considered. Which is it?

We are sorry for the confusion; nine seasons is correct as written in the manuscript.

**L38** I suggest changing "this feature" to "the CFC feature" for clarity (in my first reading of this sentence I thought "feature" refered to the bent-back front mentioned in the previous sentence).

We changed this to "CFC": "*Considering recent cases of Shapiro-Keyser cyclones (e.g., Egon 2017, Xavier 2017, Friederike 2018; see Eisenstein et al., 2022a),* **CFC** *appears to be more common following the Norwegian cyclone model (Bjerknes1919).*"

**L60** By "features" here do you mean the storm tracks over the North Atlantic and North Pacific? This could be written more clearly.

We changed this to "region": "*While the two first* **regions** *are identified for all methods, the maximum over the Mediterranean is dependent on the resolution of the used data set and methodology used.*"

**Section 2.1** It would be useful add the source of the observational data set here, as stating that the observations are only available over land (which I think is true). The source is stated in the data availability section of both parts of the paper but it would be helpful to repeat it here as well.

We added this accordingly: "*The observational data set* **provided by the German Weather Service** *includes hourly surface observations* **over land** *from 2001 to mid-2019 […].*" (l110)

**L136** I appreciate that you showed in your rebuttal letter figure FR5, a figure that used the same domain for the COSMO-REA6 data as for the observational data, for comparison with Fig. 3a–h in the paper. As you still use different domains for these two datasets in the figures shown in the paper it would be helpful to note this in the paper and say briefly why you chose to use the different rather than consistent domains.

We added the following at the end of Section 2.2: "*Note that the area shows an eastward extension and a northerly shift compared to the observational data set to include more northern regions affected by winter storms, where the observational data are sparse.*" (l138)

**Section 2.4** It would be useful to add here if there is a constraint on the minimum length (in time or space) of the cyclone tracks.

We added this information to the text: "*Cyclones must travel at least 1000 km and last for at least one day to be considered.*" (l188)

**p10** Consider combining some of the 3 short paragraphs near the end of this page.

We rearranged the paragraphs to combine the first ("*Finally, Fig. 3m-p […]*") and third ("*To examine how […]*") as they are both considering the same data, while keeping the second ("*As the overall frequencies […]*") as is.

**L374 & 401** Gentile and Gray (2023) didn't introduce the term CCBa, it was used previously by Earl et al. (2017). I don't know whether Earl et al. were the first to use it though.

Indeed, that is misleading. We added the reference to Earl et al. in the introduction and corrected it in Section 3.3.:
"*[…] Gentile and Gray (2023) distinguish between the CJ travelling against the system motion (named CCBa) and the CJ wrapping around the cyclone centre (CCBb)* **following Earl et al. (2017)**." (l91f)
"*The northern part of this patch is located in the area of the warm front and is possibly connected with the CCBa as* **discussed in Earl et al. (2017) and** *Gentile and Gray (2023).*" (l374)
"*As mentioned above, the area northeast to north of the cyclone centre, which does not overlap with any of the mesoscale features, corresponds to the CCBa as described in* **Earl et al. (2017) and** *Gentile and Gray (2023).*" (l402)

**Technical errors:**

**L95** "cause" → "have" (otherwise you are saying that winds cause winds!).

This has been changed accordingly.

**L129** I would say "at 10m" etc. (rather than "in 10m"); this applies three times in this sentence.

This has been changed accordingly.

**L163** "characteristics in other..." → "characteristics of other...".

This has been changed accordingly.

**L164** "different" → "differently".

This has been changed accordingly.

**L243 and elsewhere** To be horribly pedantic, "less" should be "fewer" here (if you can count the items you should use "fewer" whereas you would use "less" for an amount, e.g., less time).

We are sorry that we missed some. We changed this in lines 219 and 243.

**L333** "they might" → "it might" (because frequency is singular) or say "as WJs might"

This has been changed accordingly.

**L429** What does "they" refer to here? Strong winds?

It refers to the features. This has been clarified.